# Microphysiological model reveals the promise of memory-like natural killer cell immunotherapy for HIV± cancer

Jose M. Ayuso [1,2,3] ✉, Mehtab Farooqui[1,3], María Virumbrales-Muñoz [1,3,4], Katheryn Denecke[1], Shujah Rehman [5], Rebecca Schmitz[3,5,6], Jorge F. Guerrero [7,8], Cristina Sanchez-de-Diego[1,3], Sara Abizanda Campo[2,3], Elizabeth M. Maly[3,5], Matthew H. Forsberg[9], Sheena C. Kerr[1,3], Robert Striker[10,11], Nathan M. Sherer [7,8], Paul M. Harari[12], Christian M. Capitini [3,9], Melissa C. Skala [3,5,6] & David J. Beebe [1,3,6]

Numerous studies are exploring the use of cell adoptive therapies to treat hematological malignancies as well as solid tumors. However, there are numerous factors that dampen the immune response, including viruses like human immunodeficiency virus. In this study, we leverage human-derived microphysiological models to reverse-engineer the HIV-immune system interaction and evaluate the potential of memory-like natural killer cells for HIV+ head and neck cancer, one of the most common tumors in patients living with human immunodeficiency virus. Here, we evaluate multiple aspects of the memory-like natural killer cell response in human-derived bioengineered environments, including immune cell extravasation, tumor penetration, tumor killing, T cell dependence, virus suppression, and compatibility with retroviral medication. Overall, these results suggest that memory-like natural killer cells are capable of operating without T cell assistance and could simultaneously destroy head and neck cancer cells as well as reduce viral latency.

Immunotherapy is changing the landscape of cancer treatment, significantly improving patient outcomes across numerous tumor types[1]. Antibodies targeting immune checkpoint (e.g., PD-1/PD-L1, CTLA-4) have become the first-line treatment for multiple malignancies such as head and neck squamous cell carcinoma (HNSCC)[2]. Additionally, numerous clinical trials are exploring the therapeutic potential of T cell injections into the patient bloodstream (i.e., adoptive cell therapy)[3]. Similarly, researchers are also evaluating the use of adoptive cell therapy based on natural killer (NK) cells, and both approaches are showing promising results. However, immunotherapy relies on the patient's immune system to fight against the tumor, creating debate about the risks and benefits that immunocompromised patients may experience from immunotherapy[4,5]. Thus, the scientific community continues to adapt immunotherapy guidelines to ensure immunocompromised individuals also have access to the best medical care available[6]. There are numerous conditions that can lead to

[1]Department of Pathology & Laboratory Medicine, University of Wisconsin School of Medicine and Public Health, Madison, WI, USA. [2]Department of Dermatology, University of Wisconsin School of Medicine and Public Health, Madison, WI, USA. [3]The University of Wisconsin Carbone Cancer Center, University of Wisconsin, Madison, WI, USA. [4]Department of Cell and Regenerative Biology, University of Wisconsin School of Medicine and Public Health, Madison, WI, USA. [5]Morgridge Institute for Research, 330 N Orchard street, Madison, WI, USA. [6]Department of Biomedical Engineering, University of Wisconsin, Madison, WI, USA. [7]McArdle Laboratory for Cancer Research, University of Wisconsin, Madison, USA. [8]Institute for Molecular Virology, University of Wisconsin, Madison, WI, USA. [9]Department of Pediatrics, University of Wisconsin School of Medicine and Public Health, Madison, USA. [10]Department of Medicine, University of Wisconsin School of Medicine and Public Health, Madison, USA. [11]Vivent Health, Milwaukee, USA. [12]Department of Human Oncology, University of Wisconsin School of Medicine and Public Health, Madison, WI, USA. ✉e-mail: ayusodomingu@wisc.edu

immunosuppression, including infectious diseases (e.g., human immunodeficiency virus-1 [HIV-1])[4], medication (e.g., organ transplant immunosuppressants)[7], and genetic disorders[8]. Immunocompromised patients exhibit increased risk of cancer, resulting in an especially vulnerable cohort[9–11]. This situation is especially relevant for the almost 40 million people living with HIV worldwide. HIV antiretroviral therapy (ART) has increased the lifespan of people living with HIV to >40 years[12]. Thus, as people living with HIV age, their risk of cancer increases, with HNSCC ranking among the highest[13]. HIV-1 infects CD4 T cells, also known as T helper cells, which coordinate the immune response by secreting pro- or anti-inflammatory cytokines[14]. Thus, destruction of CD4 T cells compromises the capacity of the immune system to orchestrate an efficient response, leaving the patient vulnerable to opportunistic infections (e.g., human papillomavirus [HPV]) and cancer[14]. Memory-like natural killer (ML-NK) cells are a subset of NK cells that may offer multiple advantages to treat HIV+ cancer[15]: (1) ML-NK cells exhibit significantly elongated in vivo persistence compared with naive NK cells; (2) NK cells do not mediate graft versus host disease, allowing clinicians to use NK cells from healthy donors instead of relying on exhausted and often chemotherapy-treated cells from the patient; (3) naive NK cells exhibit cytotoxic activity against HIV-infected T cells, which would offer clinicians a therapy to simultaneously treat cancer and HIV-1 infection. Traditionally, NK cells have been considered part of the innate system due to a variety of reasons including their rapid activation in the absence of prior antigen exposure and the lack of somatically rearranged receptors, as observed in T cells[16]. However, in the last years, mounting scientific evidence has challenged such classification, showing that NK cells share several critical features typically ascribed to the adaptive response, such as antigen-specific clonal-like expansion; as well as the development of a persistent memory-like cell pool that exhibits strong recall responses after repeated antigen exposure[16,17]. Early studies in mouse models revealed that after exposure to mouse cytomegalovirus (MCMV), ML-NK cells persisted in the mouse's blood for weeks and even months[15]. Similarly, exposure to human CMV leads to NKG2C+ ML-NK cell expansion in humans, which are still detected in the patients' blood after a year even in the absence of detectable CMV viremia[18]. Subsequent studies demonstrated NK cells develop a ML phenotype in response to a variety of bacteria (e.g., Mycobacterium) and viruses, including influenza, HIV, CMV, Epstein–Barr, or Herpes simplex virus[19]. Previous studies have shown that CMV infection can lead to the generation of a NKG2C+ ML-NK cell subset that reduces HIV viremia during primary infection, highlighting the critical role of ML-NK cells in HIV progression[19]. On the other hand, our knowledge about the mechanisms controlling ML-NK cell distribution remains limited. Hapten-derived ML-NK cells concentrate in the liver, while influenza-specific ML NK cells are present in liver and lung[20]. Finally, cytokine-induced ML-NK cells have been detected systematically[20]. Altogether, these studies highlighted ML-NK cells play a critical role in HIV and tumor progression, suggesting therapies based on injections of allogenic ML-NK cells could provide a versatile tool for HIV+ cancer patients. In 2016, a first-in-human phase 1 clinical trial demonstrated that adoptive ML-NK cell therapy was feasible and safe in humans, leading to successful ML-NK cell proliferation and expansion in acute lymphoblastic leukemia patients[21]. This study also showed that five out of nine patients developed robust anti-tumor response against AML cells, with four of them exhibiting complete remission[21]. Another phase 1 clinical trial explored the potential of ML-NK cell therapy combined with chemotherapy in AML patients. The results showed that chemotherapy (i.e., fludarabine, cytarabine, and filgrastim) followed by ML NK cell therapy led to donor-derived ML-NK cell expansion for more than 3 months, and complete remission in four out of eight patients with no significant toxicity[22]. Other studies are exploring the use of single cell analysis techniques to develop molecular signatures (e.g., NKG2A) to identify and predict NK cell donors that will lead to optimal ML-NK cell

response against tumor cells in humans[23]. As our knowledge of ML-NK cells advances, additional clinical trials continue to report promising results regarding the potential of ML-NK cells to treat hematological cancers[24,25], hoping to expand soon these findings to solid tumors[16]. However, evaluating ML-NK cell potential for HIV+ solid tumors remain challenging due to the complexity of the tumor microenvironment and HIV-1 infection. Previous studies using HIV-1 in animals (e.g., macaques, chimpanzees, gibbons) have provided valuable lessons about HIV biology[26] and SIV research in macaques has significantly advanced our knowledge[27]. However, these models still have some limitations. HIV-1 does not lead to AIDS-like symptoms in animals, which limits the potential of in vivo models to evaluate immunotherapy responses ML-NK[28]. On the other hand, traditional in vitro models struggle to capture the tumor microenvironment[29]. Solid tumors such as HNSCC generate an immunosuppressive microenvironment characterized by hypoxia, nutrient starvation, waste product accumulation, and acidic pH, which significantly limits the efficacy of the immune system to fight the tumor[30]. This complex microenvironment is challenging to capture with traditional in vitro systems based on Petri dishes, limiting the translation of in vitro observations into clinically actionable results[31]. In this context, advances in microfluidics and microfabrication techniques have led to the development of sophisticated microphysiological systems (MPS), also known as organ-on-a-chip, that allow us to capture the tissue microstructure using human cells[32]. Recent studies are using MPS to mimic the tissue microenvironment to evaluate the efficacy of experimental immunotherapies using immune checkpoint inhibitors or adoptive cell immunotherapy[33,34]. Thus, we optimized a MPS, to culture patient derived HNSCC multicellular spheroids in a 3D collagen extracellular matrix[35] and evaluate ML-NK cell efficacy in the presence or absence of HIV-infected CD4 T cells. Our results demonstrated that allogenic ML-NK cells were able to extravasate and penetrate patient derived tumor spheroids, although the presence of CD4 T cells sped up this process. Additionally, ML-NK cells retained >80% of their cytotoxic capacity against tumor cells even in the complete absence of CD4+ T cells. Finally, we observed that ML-NK cells also detected the presence of HIV-1 infection in CD4+ T cells, potentially decreasing HIV viral protein expression.

## Results

### MPS to study HIV+ cancer

In this work, we have used an in vitro MPS to study the potential of ML-NK cells to treat HNSCC in the presence of HIV-1 infection (Fig. 1A). The MPS is made of polydimethylsiloxane (PDMS) (Fig. 1B) and includes two lateral lumens lined with human endothelial cells (HUVECs), providing us with a biomimetic vasculature to perfuse nutrients, HIV-1 virions, anti-retroviral (ART) therapy medication, and human immune cells such as primary CD4+ T cells and ML-NK cells (Fig. 1C). This approach illustrates how human-derived MPS(s) provide a versatile tool to evaluate immunotherapy for cancer patients living with HIV (Fig. 1D). We used fluorescent HIV-1 reporters and confocal microscopy to evaluate the influence of HIV-1 infection and CD4+ T cells in NK cell response. HIV-1 reporters carried a genetically modified copy of the HIV-1 genome including mCherry and yellow fluorescent protein (YFP) (Fig. 1E). mCherry expression was controlled by an early-stage promoter, signaling the early steps of the viral cycle after CD4 T cell infection. YFP synthesis required the expression of late-stage HIV transcription factor Rev, allowing us to detect the final stages of the HIV viral cycle inside of the CD4+ T cell. Overall, these dual-color HIV reporters allowed us to monitor HIV infection and progression through the viral cycle (Fig. 1E), while confocal microscopy enabled us to detect tumor killing by ML-NK cells (Fig. 1F).

### ML-NK cells to treat HNSCC

ML-NK cells represent a subset of NK cells that arise after antigen or cytokine exposure[36], and exhibit longer in vivo persistence, as well as

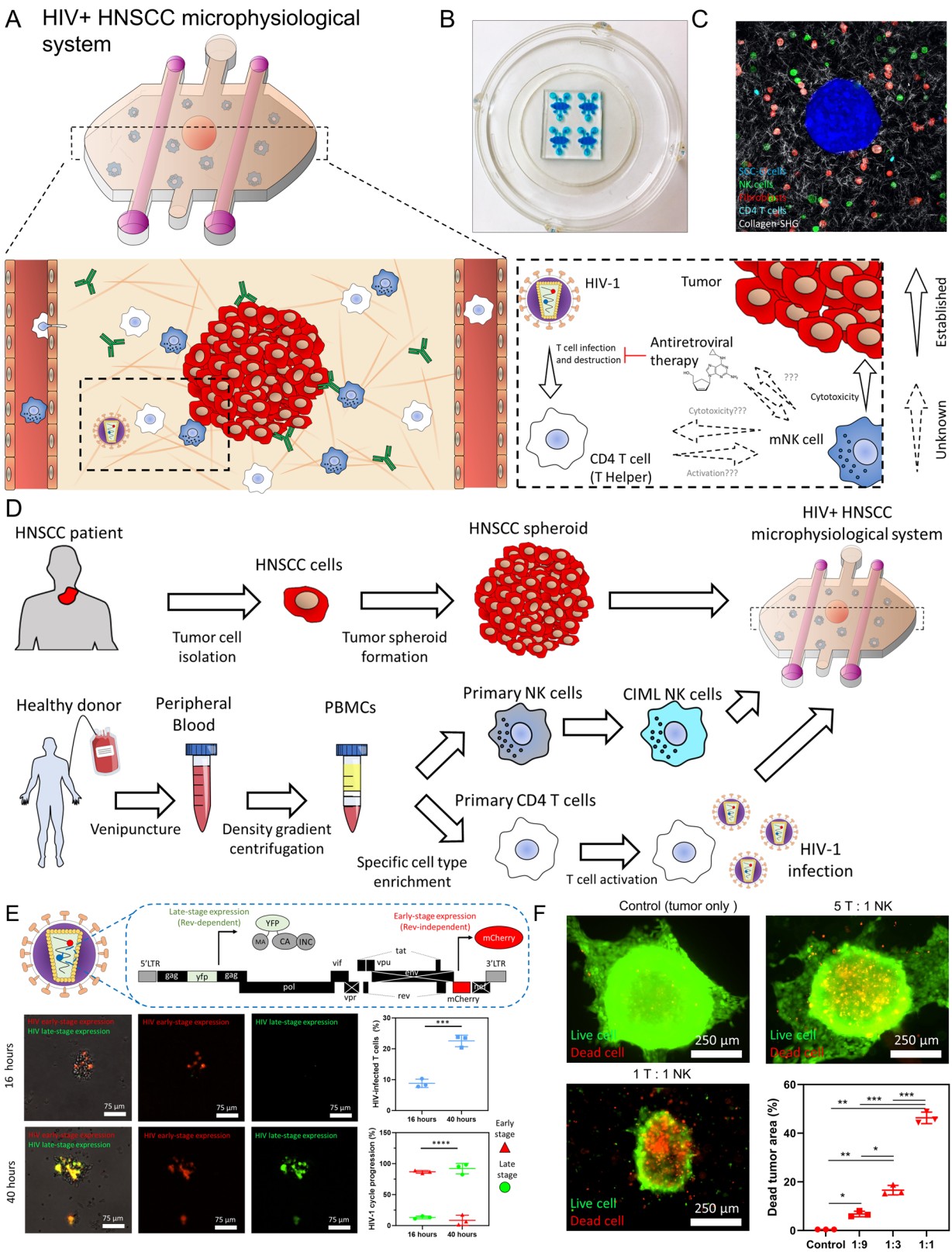

faster response to secondary cytokine or antigen exposures compared with naive NK cells[21,37]. In this work, we generated ML-NK cells by exposing primary NK cells to 10 ng/mL IL-12-, 50 ng/mL IL15, and 50 ng/mL IL-18 for 16 h followed by an additional week in culture in the presence of 1 ng/mL IL-15. Transcriptomic analyses revealed that ML-NK cells upregulated multiple genes related to NK cell survival, proliferation, and cytotoxicity (Fig. 2A–E). Our ML-NK cells exhibited

upregulation of multiple genes related with NK cell survival, proliferation, and effector function, including interleukin (IL) 2, IL15, IL-12, interferon alpha (IFNα), granzyme B (GZMB) (Fig. 2B). Among the genes analyzed, the most common alteration observed in ML-NK cells was persistent upregulation of gene expression (Fig. 2E). Next, we tested the ML-NK cell cytotoxic capacity against patient derived HNSCC cells (including the HNSCC cell line SCC-6 as a standardized

**Fig. 1 | Microphysiological model to study NK cell immunotherapy in HIV⁺ HNSCC. A** Scheme showing the structure of the microphysiological system (MPS) used. A HNSCC tumor spheroid and CD4 T cells were suspended in a 3D collagen hydrogel flanked by two lateral lumens lined with endothelial cells (i.e., HUVECs), providing a biomimetic vasculature to perfuse media and other immune cells such as natural killer (NK) cells. Schemes showing the complex interactions during NK cell immunotherapy in HIV⁺ HNSCC: (I) CD4 T cells potential modulation of NK cell response; (II) HIV-1 infection hinders CD4 T cell response; (III) NK cells can target tumor cells coated with antibodies in a CD4 T cell-independent manner; (IV) further NK cells can potentially modulate HIV-1 infection progression by engaging with HIV-infected CD4 T cells. **B** Picture of a 4-microdevice array filled with colored water for visualization purposes. **C** Confocal image showing the HNSCC spheroid, CD4 T cells, NK cells, and HNSCC fibroblasts labeled in different colors in the 3D collagen hydrogel. **D** Protocol followed to study NK cell immunotherapy response in HIV⁺ HNSCC. **E** HIV-1 infection of CD4 T cells was monitored with genetically modified single-round HIV-1 viruses encoding dual fluorescent reporters. Primary CD4 T cells infected with HIV-1 express the fluorescent protein mCherry and YFP during early and late-stage infection. Fluorescence images show a representative result. The experiment was repeated using CD4 T cells from three different blood donors, acquiring five images per experiment. Images from the same experiment were averaged and used as an independent replicate. Graphs show average of these three independent replicates ± standard deviation. *P* value was set to 0.05. **F** Cytotoxicity assay showing NK-92 cells cytotoxicity against HNSCC SCC-1 spheroids in the model at different tumor cell:NK cell ratios. The experiment was repeated three independent times, analyzing 1 spheroid per image. Graph shows the area occupied by dead tumor cells compared to the spheroid area. Bar graphs show average ± standard deviation. Data were analyzed using Brown–Forsythe and Welch ANOVA tests. *, **, ***, **** denote *P* value < 0.05, <0.01, <0.005, <0.001 respectively.

cell line) (Fig. 2F, G). The results demonstrated that ML-NK cells exhibited higher cytotoxicity compared to naive NK cells and NK cell lines (i.e., NK-92) against HNSCC cell lines as well as patient-derived cells (Fig. 2H and Supporting Figs. 1 and 2). Additionally, ML-NK cells showed significantly lower cytokine dependence (e.g., IL-12) to exert their cytotoxic functions (Fig. 2F). Interestingly, ML-NK, naive NK, and NK-92 cells exhibited a similar cytotoxicity profile against HSNCC samples (i.e., sample #CK7521 was the most vulnerable, whereas sample #CK5566 remained the most resistant). Additionally, ML-NK cells obtained from multiple donors exhibited a consistent cytotoxicity pattern against HNSCC samples (i.e., sample #CK7521 was the most vulnerable, #CK5566 was the most resistant). However, ML-NK cells from multiple donors differed in their cytotoxic potential, highlighting the role of donor heterogeneity and the importance of biomarkers that allow us to successfully predict ML-NK cell efficacy (Supporting Fig. 3). In this context, multiple studies have explored the role of activator (e.g., NKG2D) and inhibitory NK cell receptors (e.g., NKG2A) as well as immune exhaustion markers as predictive tools[38]. Interestingly, genetic screening showed that individual molecular markers such as PD-1, MICA, CEACAM-1, or HLAs struggled to predict NK cell cytotoxicity (both primary ML-NK cells and NK cell lines) against patient-derived HNSCC samples (Supporting Fig. 3), suggesting that a more holistic approach that integrates multiple markers could help us to successfully predict NK cell sensibility. Altogether, these results highlighted the challenges of predicting patient response to cell adoptive therapy purely based on molecular approaches[39], and underscore the potential of MPSs as a complementary tool to traditional molecular approaches to evaluate treatment response in a patient-specific manner[40].

## ML-NK cells detect the presence HIV infection in CD4 T cells

Once we demonstrated the superior cytotoxic capacity of ML-NK cells, we set out to evaluate their potential against HIV-infected T cells. Previous studies have raised some concerns about long-term ART treatment leading to NK cell disruption[41]. However, allogenic NK cell transplant relies on NK cells obtained from healthy donors, which have not been exposed to ART prior injection in HIV⁺ patients. Thus, allogenic ML-NK cells would be exposed to ART medication for a shorter period of time (i.e., weeks, months) compared with NK cells obtained from HIV⁺ cancer patients (years, decades). The standard of care for HIV patients today is a combination of at least two and more typically 3 antivirals that have different mechanisms of action (e.g., integrase or protease inhibitors, and nucleoside or non-nucleoside reverse transcriptase inhibitors)[42]. Thus, we cultured ML-NK cells in the presence of Abacavir (i.e., nucleoside analog reverse transcriptase inhibitor), Atazanavir (i.e., HIV protease inhibitor), and Doravirine (i.e., non-nucleoside reverse transcriptase inhibitor), individually and in combination for 7 days. We analyzed a range of concentrations consistent with those found in plasma from patients with HIV (Max concentration found in plasma: Doravirine= 10.8 μM[43]; Abacavir 1.6 μM[44], Atazanavir 1.1 μM[45]). The results demonstrated that ART drugs did not affect ML-NK cell viability, proliferation, or cytotoxic capacity against HNSCC cell lines and patient derived HNSCC samples (Fig. 3A–F). We repeated the experiment using NK-92 cells, which showed a similar response and retained their proliferative and cytotoxic potential even when treated with ART drugs for 7 days (Supporting Fig. 4). On the other hand, optical metabolic imaging revealed that ART drugs affected NK cell metabolism and decreased their redox potential (Supporting Fig. 4). Thus, we decided to explore potential changes in gene expression and cell metabolism induced by ART drugs in ML-NK cells. The transcriptomic analysis showed that the majority (>90%) of genes analyzed were not affected by these drugs. However, there were some clinically relevant genes that showed dysregulation, including CTLA-4 upregulation, which is a well-known immune exhaustion marker (Supporting Fig. 5). Overall, ART drugs seemed to have no effect on ML-NK cell proliferation, viability, and cytotoxicity, but they led to some genetic and metabolic changes that warrant close monitoring. Therefore, future studies and clinical trials could include molecular analysis in their secondary read-outs to elucidate the mechanisms driving the molecular alterations caused by ART.

Next, we evaluated the interaction between ML-NK cells and HIV-infected CD4⁺ T cells. We isolated primary CD4⁺ T cells and activated them by CD3 and CD28 stimulation. After T cell activation, we proceeded to infect them with a single-round dual fluorescent HIV-1 reporter virus (i.e., a genetically modified HIV-1 strain encoding GFP and mCherry to visualize infected cells in green and red at the early and late stages of infection respectively) (Fig. 3G). Once we confirmed the presence of HIV-1 infection within CD4⁺ T cells, we proceeded to evaluate their interactions with ML-NK cells. When we co-cultured ML-NK cells with autologous HIV-infected CD4⁺ T cells (i.e., CD4 T cells and ML-NK cells came from the same patient), we observed that ML-NK cells clustered around the HIV-infected T cells, whereas non-infected T cells remained as isolated cells (Fig. 3H, I). This result suggested that ML-NK cells were capable of detecting the presence of the virus inside of the T cells (Fig. 3H, bottom inserts show a magnification of ML-NK cell clustering around HIV-infected T cells). We also repeated the experiment culturing allogenic ML-NK cells with HIV-infected T cells (i.e., obtained from different patients). The results demonstrated a similar trend: allogenic ML-NK cells recognized HIV-infected T cells and did not cluster around non-infected T cells (Supporting Fig. 6). Next, we used an HIV-1 latency reporter that included a copy of the GFP gene in the HIV genome, which was integrated in the CD4⁺ T cell. This HIV reporter allowed us to monitor HIV-1 reactivation in CD4⁺ T cells. HIV-1 reactivation was achieved via TNF-α exposure for 24 hours, leading to HIV-1 genome expression and GFP synthesis (Supporting Fig. 7)[46]. Next, we co-cultured the latent HIV-1 reporter cells with ML-NK cells, resulting in a significant decrease in HIV-induced GFP expression, indicating that ML-NK were decreasing HIV-1 reactivation

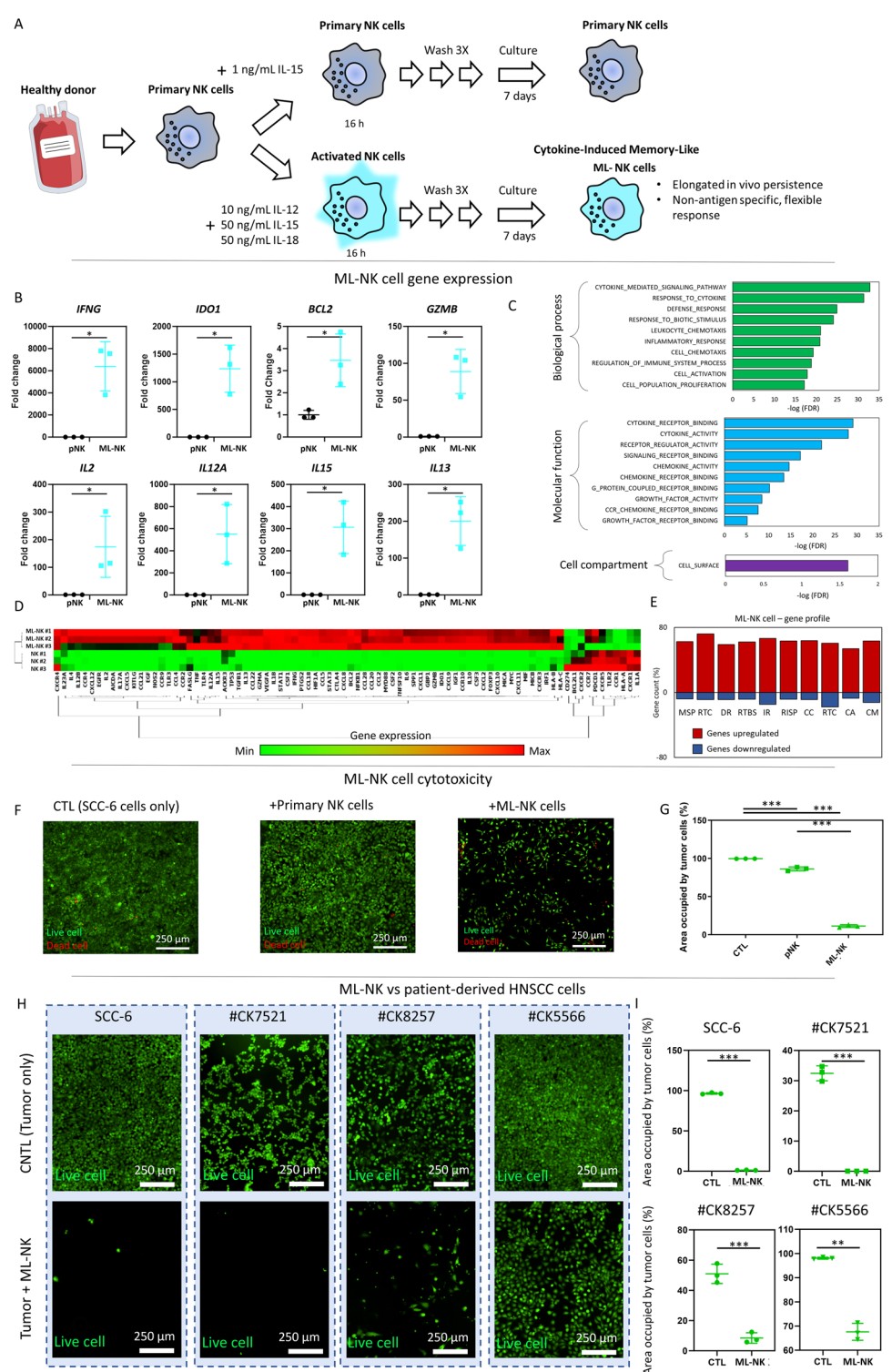

on HIV-infected T cells (Fig. 3J, K). Since ML-NK cells are generated in clinical trials by combining IL-12, IL-15, and IL-18[21], we repeated these experiments in the presence of these cytokines to test whether they could lead to undesired HIV-1 reactivation. These cytokines alone did not reactivate latent-HIV, and more importantly, they did not interfere with the capacity of ML-NK cells to suppress HIV-1 reactivation (Supporting Figs. 8 and 9). Additionally, we observed that both allogeneic as well as autologous ML-NK cells led to similar HIV-1 reactivation suppression, opening the possibility of using ML-NK cells from healthy donors, which, have never been treated with ART, chemotherapy, or radiation. Additionally, allogenic NK cells commonly exhibit increased

cytotoxicity compared with autologous NK cells due to HLA-KIR mismatches[47,48]. In summary, these results support ML-NK cell retention of cytotoxic capacity in the presence of ART drugs and they do not increase HIV-1 progression, potentially offering an off-the-shelf cellular therapy option[49–51] to explore for patients living with HIV+ cancer HIV+.

## ML-NK cells as a tool to treat HIV± HNSCC

Finally, we decided to leverage our MPS to explore whether the presence or absence of CD4+ T cells affected allogenic ML-NK cell cytotoxicity against HNSCC cells (mimicking HIV- vs HIV+ tumors). Experiments in traditional 2D systems (i.e., Petri dishes, transwells)

**Fig. 2 | ML-NK differentiation and behavior. A** Protocol describing cytokine-induced memory-like NK cell generation. Primary NK cells were isolated form healthy donors and cultured for 30 hours in NK media supplemented with IL12, IL15, and IL-18. Next, primary NK cells were cultured for 7 days to allow ML-NK cell differentiation. **B** Gene expression analysis comparing primary NK cells with ML-NK cells. ML-NK cells showed upregulation of multiple NK cell activation markers. Data were analyzed using paired *t* test. **C** Gene-ontology analysis showing biological processes (green graph), molecular function (blue graph), and cellular compartment (purple graph) affected in ML-NK cells according to the gene expression results. Data were generated using GSEA (https://www.gsea-msigdb.org/). **D** Clustergram showing differentially expressed genes in ML-NK cells compared with naive NK cells. **E** Graphs showing the percentage of genes upregulated or downregulated in the cellular functions identified in the GSEA analysis. **F, G** Primary NK and ML-NK cells were co-cultured for 24 h with the HNSCC cell line SCC-6 to analyze their cytotoxic potential against established cell lines. ML-NK cells showed increased cytotoxic capacity compared with primary NK cells. Data were analyzed using Brown-Forsythe and Welch ANOVA tests. **H, I** Images show ML-NK cell cytotoxicity against patient-derived HNSC tumor cells after 3 days in co-culture (SCC-6 are included as comparison). Fluorescence images show a representative experiment. The experiment was repeated three independent times, acquiring five images per experiment. Images from the same experiment were averaged and used as an independent replicate. Graphs show average of these three independent replicates ± standard deviation. Data were analyzed using Brown–Forsythe and Welch ANOVA tests. *P* value was set to 0.05. Bar graphs show average ± standard deviation. *, **, *** denote *P* value < 0.05, <0.01, <0.005, respectively.

revealed that CD4 T cells alone did not kill tumor cells, whereas ML-NK cells killed 85% of patient derived tumor cells in the absence of T cells (Fig. 4A). Interestingly, we observed an increase in tumor cell killing when CD4 T cells were present, (93% HNSCC cells dead) suggesting that they support ML-NK cell cytotoxicity (Fig. 4A, B). Thus, we leveraged our MPS to evaluate the effect of CD4 T cells on multiple key aspects of the ML-NK cell response, including ML-NK cell extravasation, tumor penetration, and cytotoxicity in a more physiologically relevant scenario (Fig. 4C). We observed that after 3 hours in the MPS, the presence of CD4 T cells in the matrix induced ML-NK cell extravasation from the biomimetic blood vessels, whereas in the absence of T cells, most ML-NK cells remained inside of the vessel (Fig. 4D). This result suggested that activated $CD4^+$ T cells led to ML-NK cell recruitment in the short-term. Interestingly, after 24 hours, ML-NK cells extravasated and migrated towards the tumor spheroid even in the absence of T cells, although the presence of $CD4^+$ T cells increased the number of extravasating ML-NK cells (Fig. 4E–G). Further, CD4 T cells did not increase the distance migrated by ML-NK cells toward the tumor (Fig. 4H) nor affected the number of ML-NK cells penetrating the patient derived tumor spheroid (Fig. 4I–L), showing that ML-NK cells may exhibit clinically relevant extravasation and tumor penetration even in the absence of CD4 T cells. Finally, we evaluated whether the presence of CD4 T cells is required for ML-NK cell cytotoxic capacity in 3D environments. The results demonstrated that after 24 hours, 35% of the tumor spheroid was dead in the presence of ML-NK cells alone, whereas in the co-culture with $CD4^+$ T cells 50% of the area was dead (Fig. 4I–L). Overall, these results demonstrated that the presence of CD4 T cells increased the cytotoxic potential of ML-NK cells and contributed to ML-NK cells extravasation. However, the results also demonstrated that even in the complete absence of $CD4^+$ T cells, ML-NK cells were able to extravasate, although at a slower pace, and killed 35% of the tumor mass. We observed a similar trend using the cell line NK-92, where the presence of CD4 T cells led to a moderate significant increase of NK-92 cell cytotoxicity (Supporting Fig. 11). Thus, we decided to evaluate the effect of CD4 T cells on ML-NK cell gene expression. We observed that the presence of $CD4^+$ T cells led to overexpression of several genes related to NK cell survival and function (e.g., IL-15, GZMB) (Supporting Fig. 12). Therefore, this approach could help to identify the specific signals secreted by $CD4^+$ T cells that moderately increased ML-NK cell response, leading to new therapeutic protocols where these T cell-derived signals are provided ex vivo for people living with HIV.

## Discussion

Advances in immunology have moved immunotherapy to the first line treatment in many types of cancer. However, we still do not completely understand whether immunocompromised people would experience the same benefits from adoptive cell therapy. Thus, in this study, we used an $HIV^+$ HNSCC MPS to complement traditional in vitro and animal preclinical models in the field of adoptive cell therapy. As described, previous studies have demonstrated that antigen exposure leads to the generation of recall responses by NK cells that are considered a bona fide memory response. In this study, all the ML-NK cells used were generated by exposing naive NK cells to a specific cocktail of cytokines, leading to the generation of an antigen-agnostic, memory-like phenotype. This study evaluates human ML-NK cells against patient derived HNSCC cells in the presence/absence of HIV-infected/non-infected T cells. Here, our MPS provided a robust platform to evaluate multiple clinically relevant read-outs including extravasation, tumor penetration, and cytotoxicity. Previous studies have demonstrated that after decades of treatment with ART medication, the patient's NK cells may show signs of exhaustion or impairment. Our preclinical results demonstrated that CDC-recommended ART drugs do not affect allogenic ML-NK cell viability, proliferation, or cytotoxicity in the short term, highlighting the potential of using ML-NK cells from healthy donors as an alternative cell source. Additionally, ML-NK cells limited HIV-1 activation in the model, which is consistent with recent studies showing that naive NK cells reduce HIV viremia in humanized mouse models. Finally, we showed that even in the complete absence of CD4 T cells, ML-NK cells hold promising therapeutic potential. The observation that CD4 T cells increase ML-NK cell cytotoxicity using MPSs may inform clinical trials evaluating immunotherapy, where people living with HIV could be stratified into different arms of the trial based on their levels of CD4 T cells. Since NK cells do not mediate graft versus host disease, they may offer an interesting alternative to T cells for allogenic transplants. Thus, in this project we focused on allogenic ML-NK cells as a potential therapy for $HIV^+$ HNSCC. Our results showed that allogenic ML-NK cells clustered around HIV-infected T cells, while no ML-NK cells were observed around non-infected T cells. When we repeated the experiment using autologous ML-NK and HIV-infected T cells, we observed a similar trend (i.e., autologous ML-NK cells clustered around HIV-infected T cells) (Supporting Fig. 6). One of the limitations of our study is the absence of a direct comparison between $HPV^+$ and $HPV^-$ HNSCC since all our samples were negative for HPV, HHV, Epstein-Barr, or HIV. HPV infection plays a critical role in HNSCC progression and treatment response[52]. $HPV^+$ HNSCC shows higher infiltration of cytotoxic NK cells ($CD56^{dim}$). Additionally, $HPV^+$ HNSCC exhibits lower HLA class I expression, which include ligands for several inhibitory NK cell receptors (e.g., NKG2A). In this context, low HLA class I expression is associated with better prognosis for $HPV^+$ HNSCC, but worse outcome in $HPV^-$ tumors; suggesting HPV infection contributes to the NK cell response against $HPV^+$ HNSCC. In this study, we analyzed a small number of NK cell donors. Although we tested a limited pool of donors, our results showed that ML-NK cells obtained from different donors exhibited significant differences in cytotoxic potential (Supporting Fig. 3). Factors such as the repertoire and level of expression of NKG2 receptors expressed in ML-NK cells could have a deep impact on cytotoxicity against tumor cells. Further studies may also analyze whether patient-variability also leads to differences in ML-NK cell resistance to hypoxic and nutrient-depleted environments. We believe that future studies evaluating ML-NK cell inter-donor variation across dozens, or hundreds of donors could help to define a molecular

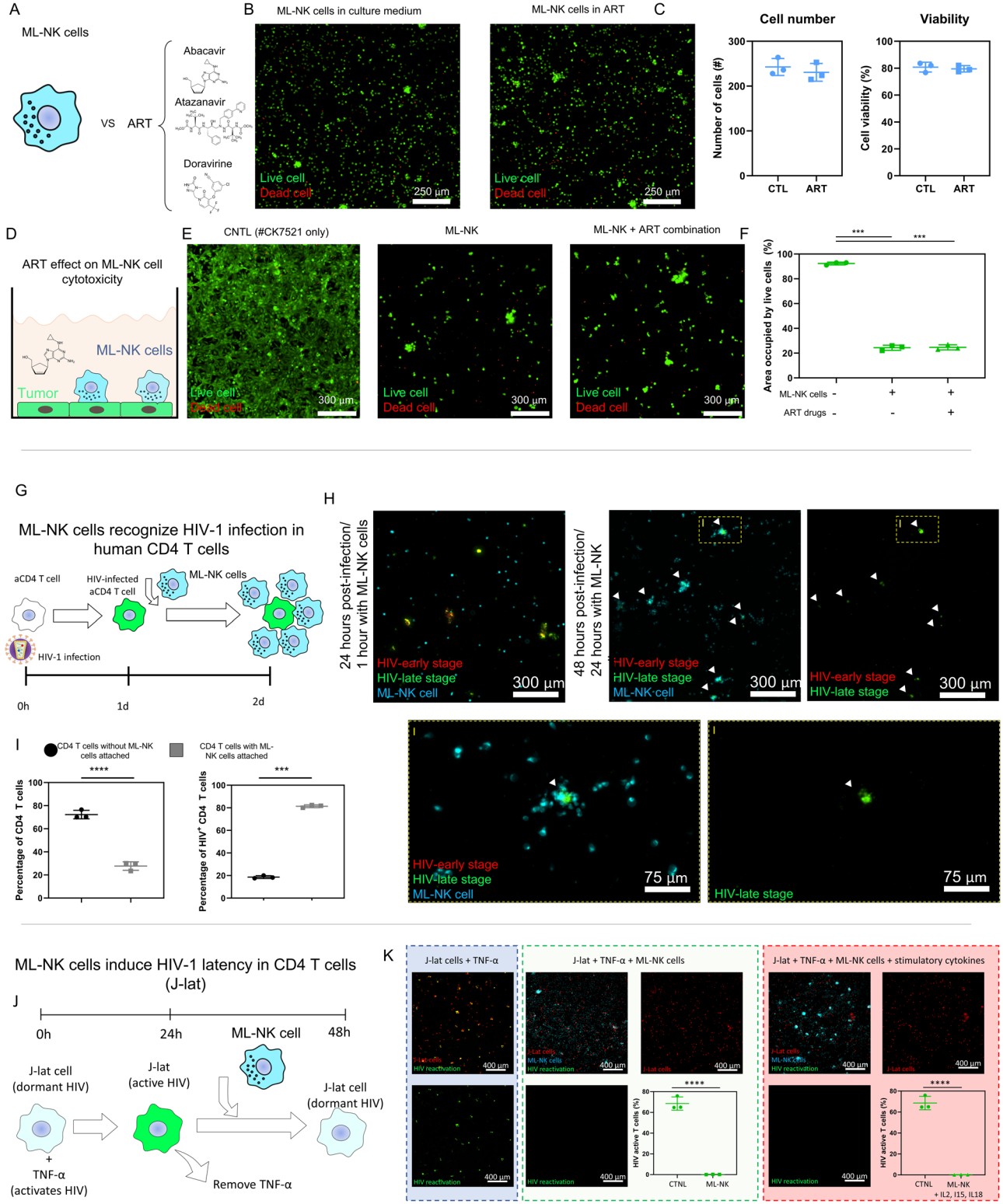

signature to optimize treatment response in a patient specific manner. While in this work we focused on HIV-1, we believe that MPS models provide a versatile complement to traditional preclinical models (i.e., 2D, animal). This study exclusively relies on in vitro systems, and we have limited our study to a few critical parameters involved in HIV-cancer-immune system interactions. Thus, additional studies combining MPS with humanized animal models would enable us to create an improved pipeline to test new treatments (e.g., ML-NK cells) across different types of immunosuppression (e.g., organ transplant patients) to provide a more robust menu of therapeutic options for challenging patients.

## Methods

Our research complies with all relevant ethical regulations and experimental protocols have been approved by the Office of Biological Safety (OBS) at UW-Madison. The study was also approved by the Heath Sciences Institutional Review Board (HS IRB, #ID: 2020-1623) at UW-Madison (staff reviewer: Rebecca Hoffman).

**Fig. 3 | ML-NK cells retain their cytotoxic capacity in the presence of ART drugs and reduce HIV-1 activation. A** We cultured ML-NK cells in the presence of the ART drugs Abacavir (nucleoside analog reverse transcriptase inhibitor), Atazanavir (protease inhibitor), Doravirine (non-nucleoside reverse transcriptase inhibitor). **B, C** Microscopic images show viable and dead ML-NK cells stained in green and red respectively. ML-NK cells were exposed to the ART drug cocktail (10 μM Doravirine, Abacavir, and Atazanavir) for a week before analyzing viability. The bar graphs show that ART had no effect on ML-NK cell viability or cell proliferation. **D–F** ML-NK cells were cultured in the presence of the ART drug cocktail for a week and then their cytotoxicity was analyzed by adding them on top of a patient-derived HNSCC cell monolayer (#CK7521). After 24 h, ML-NK cells and floating HNSCC cells were washed, and the attached HNSCC cells were stained (green = viable cells, red = dead cells). The results demonstrate that the ART drug cocktail (10 μM Doravirine, Abacavir, and Atazanavir) did not impact ML-NK cell cytotoxicity. Experiments shown in **A–F** were performed in 96-well plates. Data were analyzed using a extra sum-of-squares *F* test to evaluate whether the different conditions differed from the control. **G–I** ML-NK cells recognize HIV infection in CD4 T cells. Primary CD4 T cells were infected with a single-round (Env-minus) HIV dual-color fluorescent reporter and cultured for 24 h to allow the virus to reach the late stage of the viral cycle (detected by GFP expression on CD4 T cells). Next, HIV-infected CD4 T cells were co-cultured with ML-NK cells (in cyan). Microscopy images shown in H demonstrate that ML-NK cells clustered around the HIV-infected CD4 T cells (Right

panels and bottom insets), whereas most of the non-infected T cells remained as single cells. Arrows indicate the location of HIV-infected CD4 T cells (labeled in green), showing most T cells had ML-NK cells in contact. Graphs in **I** show the percentage of T cells with (in white) and without (in gray) ML-NK cells attached in the absence (left graph) presence (right panel) of HIV infection. Experiments shown in **G–I** were performed in 96 well-plates. Brown-Forsythe and Welch ANOVA tests. **J, K** We used an HIV latency model (i.e., J-lat10.6 cells) to study the potential effects of ML-NK cells on HIV latency. First, we induced HIV reactivation in J-lat cells by culturing them in the presence of TNF-α for 24 h. HIV reactivation was detected by GFP expression in J-lat cells. Next, J-lat cells were co-cultured with ML-NK cells. After an additional 24 h, we observed that ML-NK cells led to a significant decrease in the number of GFP + J-lat cells, suggesting suppression of HIV-1 reactivation. ML-NK cells exhibit a similar effect regardless of the presence or absence of stimulatory cytokines. Experiments shown in **G–K** were performed in the MPS. Autologous primary ML-NK and CD4 T cells were used in these experiments. Confocal images show a representative result. The experiments were repeated three independent times, acquiring five images per experiment. Images from the same experiment were averaged and used as an independent replicate. Graphs show average of these three independent replicates ± standard deviation. Data were analyzed using unpaired *t* test with Welch's correction. *P* value was set to 0.05. ***, **** denote *P* value < 0.005 and 0.001 respectively.

## Reagents

IL-2, IL-12, IL-15, and IL-21 were purchased from Peprotech, whereas IL-18 was acquired from R&D. 50 μg 50 μg of IL-2 was diluted in 500 μL of 50 mM Acetic acid with 0.1%BSA (100 ng/mL). The other cytokines were reconstituted in PBS at 100 μg/mL.

## Microdevice design and fabrication

Microdevice fabrication is described in more detail in ref. 23. Briefly, the template was designed in illustrator and fabricated using SU-8 based lithography. Next, PDMS-based microdevices were fabricated using the SU-8 template and the microdevices were treated with oxygen plasma and bonded to a 60 mm glass bottom Petri dish. Microdevices were sterilized by UV exposure for 15 min prior to cell culture. The final microdevice was comprised of a central microchamber to inject a 3D hydrogel and three parallel 340 μm-diameter PDMS rods. To increase hydrogel attachment to the PDMS, the microdevices were treated 10 min with poly(ethyleneimine) (Sigma-Aldrich, 03880) diluted at 1%; followed by 30 min with glutaraldehyde (Sigma-Aldrich, G6257) diluted at 0.1% in water; and finally washed three times with water.

## Cell culture

HNSCC cell line SCC-6 (SCC073, Millipore-Sigma) was cultured in RPMI 1640 (Lonza) supplemented with 10% FBS (VWR), 2 mM L-Glutamine (Thermo Fisher), and Penicillin/Streptomycin (Thermo Fisher). HNSCC surgical samples were collected from the UW-biobank and cut into <1 mm pieces using a surgical blade. Tumor fragments were digested in RPMI-1640 supplemented with collagenase 1 (8 mg/mL), collagenase 4 (4 mg/mL), hyaluronidase (4 mg/mL), and DNAse I 1 mg/mL (all obtained from Worthington Biochemical Corporation) for 4 h at 37 °C in continuous agitation. Next, the tissue/cell solution was centrifuged at 400 × g for 5 min, the pellet was resuspended in PBS and filtered through a sterile 100-μm filter to remove ECM fragments. Finally, the filtered cell suspension with was cultured in Advanced DMEM/F-12 (Invitrogen), supplemented with 5% FBS, 0.4 ng/mL hydrocortisone (Sigma), 10 ng/mL EGF (Invitrogen), 24 μg/mL adenine in 1 mM HCl, Penicillin/Streptomycin, 2 mM L-Glutamine, and 10 μM Y-27632 dihydrochloride. NK-92 cells (NK-92® - CRL-2407, ATCC) were cultured in X-VIVO 15 (Lonza) supplemented with 20% FBS (VWR) and 100 U/mL IL-2. J-lat cells were cultured in RPMI Medium 1640 (1X) + L-Glutamine supplemented with 10% FBS (VWR) and 1% Penicillin/Streptomycin and HIV-1 expression was induced by exposing J-lat cells to 10 ng/mL TNF-α. Samples included in this study were negative for HIV, HPV, HHV, EBV.

## Spheroid generation

Tumor spheroids were generated by the hanging drop method. Briefly, MCF7 cells were trypsinized and resuspended at 40 cells/mL in media supplemented with 20% 12 g/L methylcellulose dissolved in media. 25 μL droplets were placed on top of a Petri dish lid and distilled water was added to the bottom of the dish to reduce evaporation during the spheroid formation. After 2 days in the incubator, one single spheroid per droplet was formed.

## Primary NK cell isolation

Primary NK cells were isolated from 30 mL whole blood obtained from UW biobank through the required informed consent forms. 30 mL of blood was separated into two 50 mL conical tubes with 15 mL of blood in each and incubated with the RosetteSep Human NK cell enrichment cocktail (Stem Cell, 15065) was added to each sample at 50 μL/mL and incubated at room temperature for 10 min. Blood was diluted 1:1 with PBS and centrifuged for 10 min at 1200 × g in sepMate tubes with 15 mL of lymphoprep at the bottom. The supernatant, containing the NK cells was pour into a new standard 50 mL tube. Red blood cells were removed by incubating the cell suspension in red blood lysis buffer for 10 min, followed by 2 additional washing steps (5 min, 400 × g).

## Cytokine-induced memory-like NK (ML-NK) cells

To generate ML-NK cells, primary NK cells were briefly stimulated with cytokines and then cultured for a week in low cytokine media to induce cell differentiation. More specifically, primary NK cells were cultured in X-VIVO15 with 10% human AB serum supplemented 10 ng/mL Il-12, 50 ng/mL IL-15, and 50 ng/mL IL-18 for 20 h. Next, cytokines were removed by washing the NK cell suspension three times with PBS (300 × g for 5 min). NK cells were cultured at 3 × 10^6 cells/mL for 7 days in X-VIVO15 with 10% human AB serum and 1 ng/mL IL-15 to promote survival and allow differentiation in ML-NK cells. ML-NK cells generated in this study were obtained from approximately 20 different healthy donors through the UW-Madison biobank.

## CD4 T cell isolation and activation

Primary CD4 T cells were isolated from 15 mL whole blood obtained from UW biobank through the required informed consent forms. First, 15 mL of blood was diluted 1:1 in PBS and then centrifuged on Sepmate-50 tubes (StemCell) containing 15 mL lymphoprep (StemCell) at the bottom. Sepmate-50 tubes were centrifuged for 10 min at 1200 × g to separate peripheral blood mononuclear cells (PBMCs). Next, CD4

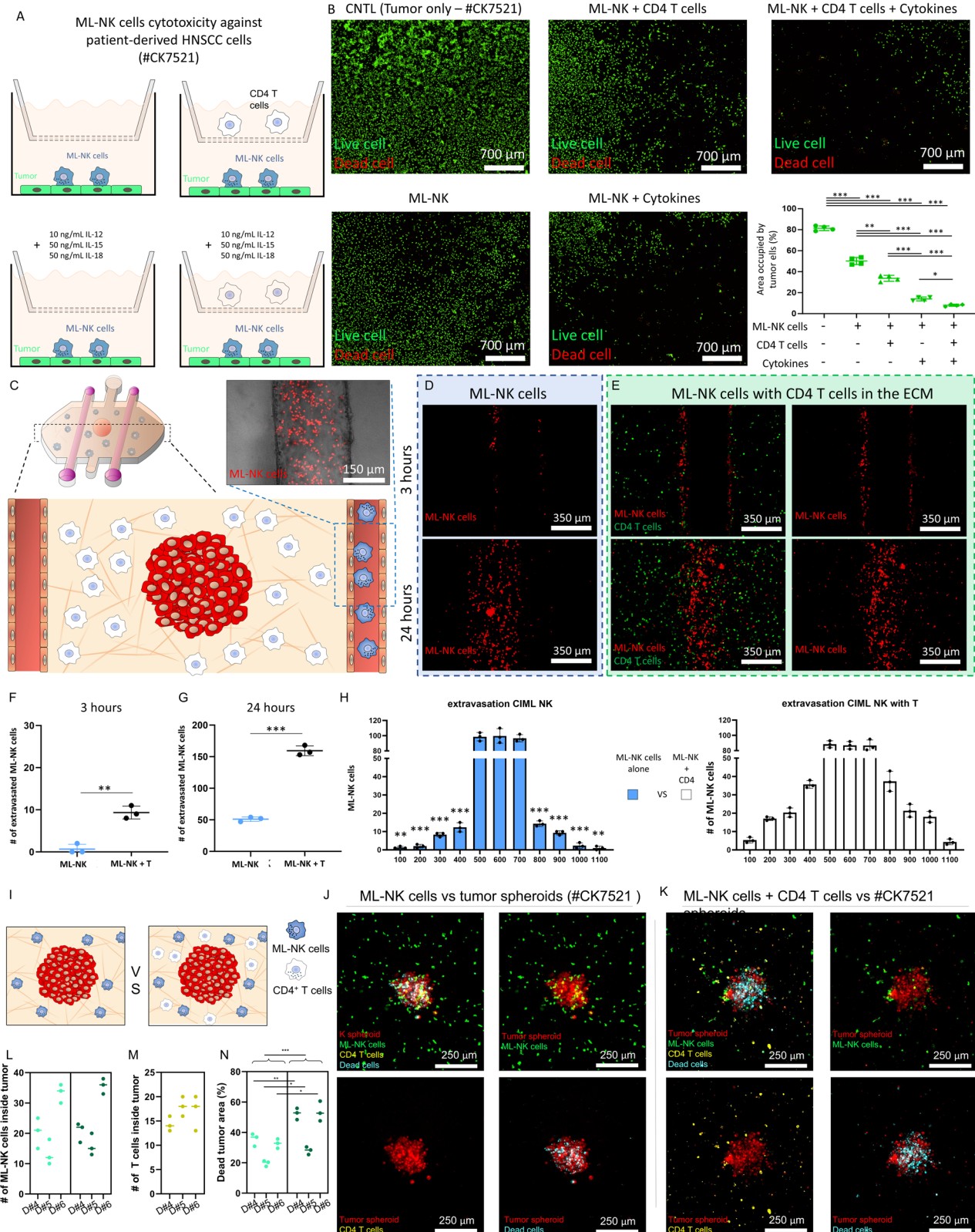

T cells were obtained by negative selection using a "CD4 + T Cell Iso-lation Kit, human" kit (miltenyi) and following the manufacturer instructions. Briefly, non-CD4 T cells were labeled with magnetic beads and removed from the cell suspension by magnetic separation, leaving a >95% enriched CD4 T cell suspension. If T cell activation was required, CD4 T cells were cultured in RPMI 1640 supplemented with 10% FBS, 40 U/mL IL-2, and anti-CD3/CD28 beads (Dynabeads™ Human

T-Activator CD3/CD28 for T Cell Expansion and Activation) used fol-lowing the manufacturer's instructions.

## HIV-1 reporter virus generation
In all, $2.5 \times 10^6$ HEK293T cells were plated in 10 cm tissue-culture-treated dishes in 10 mL media, then transfected using poly-ethylenimine with 1 μg of DNA plasmid expressing the G envelope

**Fig. 4 | Effect of activated CD4 T cells in ML-NK cells. A** Effect of activated CD4 T cells on ML-NK cell cytotoxicity against patient-derived HNSCC cells (#CK7521). ML-NK cells were seeded on top of patient-derived HNSCC cells, whereas activated CD4 T cells were added to the top insert. **B** Microscopic images showing live (Calcein AM) and dead (PI) patient-derived HNSCC after 3 days in culture. The bar graph indicates the area occupied by live patient-derived HNSCC in the different conditions. Experiments shown in **A**, **B** were performed in 24 transwell plates. Data were analyzed using Brown–Forsythe and Welch ANOVA tests and Dunnett's T3 multiple comparisons test. **C** Effect of activated CD4 T cells on M extravasation. Activated CD4 T cells were stained in green (Vybrant DiD) and embedded in the collagen hydrogel, whereas ML-NK cells were stained in red (Vybrant DiL) and perfused through the lumen. **D**, **E** After 3 and 24 h, MPS were imaged visualized by confocal microscopy to visualize ML-NK cell extravasation. **F**, **G** Bar graphs show the number of ML-NK cells that extravasated from the lumen after 3 (**F**) and 24 h (**G**). The results showed that the presence of activated CD4 T cells in the matrix led to an increase in ML-NK cell extravasation. **H** Bar graphs show the ML-NK cell distribution histogram after 24 h across the images shown in (**D**, **E**). Images where vertically divided in multiple 100 μm-width segments and the number of ML-NK cells per section was quantified. **I** Effect of activated CD4 T cells in ML-NK cell cytotoxicity against patient-derived HNSCC spheroids (#CK7521). Data were analyzed using unpaired *t* test with Welch's correction. **J** Confocal images show ML-NK cells and patient-derived HNSCC spheroids labeled in green (Vybrant DiO) and red (Vybrant DiL), respectively. Apoptotic cells were stained in cyan. **K** Confocal images show similar experiments including activated CD4 T cells labeled in yellow. **L** The bar graph shows the number of ML-NK cells that penetrated into the tumor spheroid after 24 h in co-culture. **M** Similar graph analyzing infiltrating activated CD4 T cells. **N** Area occupied by dead cells in the tumor spheroid. Experiments shown in panels C-L were performed in the MPS. Allogenic primary ML-NK and CD4 T cells were used in these experiments. Confocal images show a representative result. The experiments were repeated using three independent donors (labeled as D#4, D#5, and D#6, respectively), acquiring three images per experiment. Images from the same experiment were averaged and used as an independent replicate. Graphs show average of these three independent replicates ± standard deviation. Data were analyzed using Brown–Forsythe and Welch ANOVA tests and Dunnett's T3 multiple comparisons test. *P* value was set to 0.05. Bar graphs show average ± standard deviation. *, **, *** denote *P* value < 0.05, <0.01, <0.005, respectively.

glycoprotein from vesicular stomatitis virus (VSV-G) and 4.5 μg lentiviral packaging plasmid psPax2 (a kind gift from Dr. Didier Trono; plasmid #12260, Addgene) and 4.5 μg plasmid DNA encoding a full-length two-color reporter NL4-3 molecular clone of HIV-1 bearing inactivating mutations in env, vpr, and expressing mCherry in the nef ORF and YFP between the matrix and capsid ORFs of Gag (E-R- Gag-YFP mCherry/nef), similar to a 2-color reporter virus described in (Knoener et al., 2017). At 24 h post-transfection, media was replaced with 4 mL fresh media. At 48 h post-transfection, culture supernatants were harvested, filtered through a sterile 0.45 μm syringe filter, and frozen at −80 °C.

### CD4 T cell infection with HIV-1 reporter
In all, $2 \times 10^5$ cells were pelleted at $200 \times g$ for 10 min within a 1.5 mL Eppendorf tube, and the media was replaced with 1 mL viral inoculum. 1 μg of polybrene was added to the tube, cells were resuspended, and the cells were centrifuged at $1200 \times g$ for 2 h at 37 °C. Cells were then resuspended in fresh media and added to microfluidic device after confirming infection through mCherry expression.

### Microdevice culture
In order to embed the spheroid/NK cells in the microdevice, a 4.0 mg/mL collagen hydrogel was prepared and injected within the central microchamber. The hydrogel was prepared as follows: 5 μL of 10× PBS, 1.12 μL of 1 N NaOH; 45 μL of 8.90 mg/mL collagen type I; and 48.88 μL of culture media w/wo patient-derived HNSCC and/or primary CD4 T cells depending on the experiment. Collagen was polymerized at room temperature for 20 min to allow the formation of longer collagen fibers. Next, PDMS rods were removed, generating a cylindrical tunnel through the collagen hydrogel (i.e., lumen). Endothelial cells (HUVECs) were injected through the lateral lumens at 20 million cells/mL to mimic the vasculature. Depending on the experiment, ML-NK cells or NK-92 cells at different concentrations and/or antibodies were embedded in the matrix or perfused through the lateral blood.

### ML-NK cell retrieval from the microdevice
Microdevices could be disassembled to expose the collagen hydrogel. Using 8 mg/mL collagenase type I, the collagen hydrogel was degraded, releasing the cells embedded in the hydrogel. Next, the cell suspension was incubated with anti-CD56 magnetic beads (miltenyi) to isolate ML-NK cells following the supplier's protocol.

### Gene expression analysis by RT-qPCR array
To study how NK-92, naive NK, and ML-NK cells adapted gene expression to the different conditions evaluated (e.g., cultured in the presence of CD4 T cells, difference between naive vs ML-NK cell), we used RT-qPCR arrays from Qiagen (RT2 Profiler, PAHS-181Z, Qiagen). These arrays allow the simultaneous analysis of up to 384 genes related with immune activation/exhaustion. After isolating the desired cell type (e.g., NL-NK cells), cells were lysed, and mRNA was captured using the Dynabead mRNA DIRECT Purification Kit (Thermo Fisher Scientific, 61011). Isolated mRNA was quantified using a Qubit fluorometer (Thermo Fisher Scientific) and the Qubit RNA BR Assay Kit (Q10210, Thermo Fisher Scientific). mRNA was reverse transcribed to complementary DNA (cDNA) using the RT2 PreAMP cDNA Synthesis Kit (Qiagen, 330451). cDNA quantity was analyzed by RT-qPCR using the "Cancer Inflammation and Immunity Crosstalk" Qiagen RT2 profiler panel (Qiagen, PAHS-181Z), and data were analyzed using the Qiagen online soft-ware (https://geneglobe.qiagen.com/us/analyze/) (Supporting Figure 5). These panels include several controls that are also analyzed by the Qiagen software, ensuring sample quality (e.g., positive control, retrotranscription control, contamination control) and only samples that passed all these tests were used in the analysis. Clustergrams and volcano plots were generated using Qiagen's software, bar graphs were generated in GraphPad 9.0, and Gene Set Enrichment Analysis (GSEA) graphs were generated through (https://www.gsea-msigdb.org/gsea/index.jsp).

### Confocal microscopy and image analysis
Cells were visualized in a Leica SP8 3X STED super-resolution confocal microscope equipped with a super-continuum white-light laser for fluorescent excitation from 470 nm to 670 nm, a separate 405 nm diode laser. The unit is equipped with 3 PMTs and 2 high-sensitive HyD detectors for image collection. For live-cell experiments, a stage top incubator set at 37 °C and 5% $CO_2$ was used. For NK-92, naive NK, and ML-NK cell experiments analyzing NK cell cytotoxicity, tumor penetration, extravasation, and HIV detection, five technical replicates were acquired per condition, and experiments with primary cells were repeated at least with three different donors. We used a Leica microscope equipped with an automated stage and tile-scan capabilities. This allowed us to acquire multiple ×10 images and merge them together to generate one image that covered a large area while preserving high image resolution at single cell level. To show inter-donor variability, data obtained from technical replicates were averaged to obtain the value of a given donor or condition. For antibody diffusion experiments, images were taken every 5 min. Z-stacks were performed using a 1 and 10 μm step at ×40 and ×10, respectively. NK-92 cell migration was analyzed using an automatic cell tracking plugin (TrackMate) running on Fiji® (https://fiji.sc/). A Laplacian of Gaussian filter was applied to identify the NK-92 cells and cell estimated size and threshold were set at 10 μm and 0.9, respectively.

## Optical metabolic imaging

As in prior work[53,54], autofluorescence from NAD(P)H and FAD were imaged on a two-photon microscope using a ×20 objective (Zeiss, NA = 1.0) at 750 nm excitation (400–480 nm emission) and 890 nm excitation (500–600 nm emission), respectively. The redox ratio is the fluorescence intensity of NAD(P)H divided by that of FAD, and the mean fluorescence lifetime (tm) is the weighted average of the short and long NAD(P)H lifetime components.

## Statistical analysis

Data were analyzed with Graphpad 10. Statistical significance was set at $P < 0.05$. For parametric comparisons, we used Brown-Forsythe and Welch ANOVA tests, unpaired $t$ test with Welch's correction, and paired $t$ tests. For non-parametric comparisons, a Kruskal–Wallis test was performed followed by the Mann–Whitney $U$-test. For grouped analysis ordinary Two-way ANOVA was performed. For subpopulation analysis, single-cell redox ratios were binned to generate frequency histograms representing naive and exposed NK cell redox ratio distributions. Gaussian curves were fit to the distribution data to reveal underlying populations with distinct redox ratio. The Akaike information criterion was used to assess fit quality and determine the optimal number of fitted Gaussians. All experiments were repeated at least three times and T and NK cells were obtained from >20 different donors.

## Data availability

The data presented in this study are available in the manuscript file and the Supplementary Information file. If additional data are required, contact the corresponding author. Source data are provided with this paper.

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

## Acknowledgements

This work was supported by the University of Wisconsin Carbone Cancer Center [AAB7173] (D.J.B.); NIH grants R01 CA164492, R01 CA185747 (M.C.S.), R01 CA205101 (M.C.S.), R01AI110221 (N.M.S.), P30CA014520 (Howard Bailey); and NSF grant CBET-1642287 (M.C.S.).

## Author contributions

J.M.A., M.V.-M., M.F., S.R., K.D., R.S., J.F.G., S.A.C., C.S.-d.-D., E.M.M., and M.H.F. performed the experiments and generated the experimental data. J.M.A., M.V.-M., S.R., M.F., S.C.K., N.M.S., and M.C.S. analyzed the data and generated the figures. J.M.A., M.V.-M., D.J.B., P.M.H., R.S., S.C.K., M.C.S., C.M.C., and N.M.S. interpreted the experimental observations. All the authors contributed to manuscript conception, writing, and revision.

## Competing interests

D.J.B. has an equity interest in Tasso, Inc. D.J.B. is a stockowner of Bellbrook Labs, LLC, he is also a founder, stockowner, and manager of Salus Discovery LLC, CSO Flambeau Diagnostics LLC, stockowner of Lynx Biosciences, LLC, Onexio Biosystems, LLC, Stacks to the Future, LLC, and Turba LLC, a company developing antibiotic testing devices. The other authors declare no competing interests.
