## [Peer Review File · Nature Communications]

Microphysiological model reveals the promise of memory-like
Natural Killer cell immunotherapy for HIV+ cancerREVIEWER COMMENTS

Reviewer #1 (Remarks to the Author):

The manuscript by Ayuso et al describes studies involving use of an in vitro microphysiological system (MPS) to evaluate the functionality of human cytokine-induced memory-like NK cells against HNSCC tumor cells in the presence or absence of human HIV-infected CD4+ T cells. This MPS platform represents a novel and potentially powerful tool to study NK cell-mediated defense against tumors in settings of HIV infection, which is currently limited by the lack of a small animal models of HIV and HIV-related cancers that recapitulate aspects of human disease. In the MPS system, NK cells (and HIV-infected CD4+ T cells) traffic out of endothelium-lined channels and into a tumor-containing tissue-like microenvironment where the NK cells mediate cytolysis of the tumor cells. Overall, this is an exciting study that is expected to appeal to a broad audience within the immunology field, in part due to the potential utility of the platform in addressing other open questions in the field in addition to those evaluated in this particular study.

Nevertheless, several important concerns were noted. Most importantly, it is difficult to discern whether many of the key findings are reproducible within and across donors because little or no information is provided regarding the number of donors evaluated or the number of experimental replicates for each assay. Inter-donor variation in outcomes is a concern and high variation could seriously limit the impact of the findings. For all experiments, the number of technical replicates and intra- and inter-donor replicates should be clearly stated within each figure legend and results using cells from multiple donors should be shown so that inter-donor variability can be assessed. Similarly, many critical experimental details are omitted in the manuscript, which makes it broadly difficult to assess the rigor of several experiments. These include the following:

- For all image-based data, a clear and detailed description of the methods used to analyze the images, including the number of images/field/MPS replicates evaluated for summary data, must be included.
- It is unclear what type of transcriptomic analyses (e.g. Fig. 2, Suppl Fig. 5) were performed for this study. The method (i.e. RNAseq) should be stated, the methods used for data analyses must be described, and the raw data should be made available.
- Data in Figs. 2B, D, and E would be best conveyed as a heat map showing relative expression with individual up- and down-regulated genes identified. Also, it is unclear what the yellow color in Fig 2D signifies.
- Page 6, please state the actual range of concentrations found in plasma from HIV patients so that concordance with the assay conditions can be better evaluated.
- For all subpanels in Fig. 3 and 4: please clarify/state in legend (1) which culture system was used for these studies - i.e. MPS shown in Fig.1 or another system; (2) whether all NK plus T cell co-cultures utilized autologous cells; and (3) whether NK-92 or primary NK cells were used.
- Fig. 3H: Please indicate what does each column represents and what the arrows in images represent.

- Fig. 3I: Please clarify the figure legends and X axis labels, which are small and confusing.
- Supporting Fig. 3A: Primary data should be shown, not "+, ++, etc." designation, which is not quantitative.
- Supporting Fig. 3B: Describe how fold change was calculated
- Differences in outcomes between autologous and allogeneic NK and CD4 T cell co-culture systems should be evaluated and quantified.

Other minor issues:

- The abbreviation for "memory-like NK cells" should be "ML-NK" cells, not "mNK" cells. The latter is often used as an abbreviation for mature NK cells.
- Define PDMS at first instance.
- When discussing memory-like NK responses in the text, the authors should be careful to discriminate between antigen-specific/-dependent recall responses by NK cells, which are broadly considered bona fide memory responses, and cytokine-induced memory-like responses that are antigen non-specific.
- A figure should be referenced after this sentence: "Additionally, mNK cells showed significantly lower cytokine-dependence (e.g., IL-12) to proliferate and exert their cytotoxic function".
- Fig. 3A and Supporting Fig 4. Please replace "vs" with "plus" or "+/-", since the comparison is NKs with or without ART, not mKs vs ART.
- Page 9: Supporting Figures numbers are mislabeled in the text
- Fig. 4A: Please label T cells in diagram
- Please correct typos throughout the manuscript.

Reviewer #2 (Remarks to the Author):

Human-derived microphysiological models reveal the promise of memory-like NK cell immunotherapy for patients with HIV+ cancer

Metabolism

Can we share the cancer immunotherapy revolution with HIV patients having cancer. Investigators leveraged human-derived microphysiological models to reverse-engineer the HIV-immune system interaction and evaluated the potential of memory-like natural killer (mNK) cells for HIV head and neck cancer, a non-AIDS cancer related to tobacco or HPV. mNK cells are different as they have strong cytotoxicity and activation, and live long-term. Different mNK cell response in human-derived bioengineered HIV+ environments, including mNK cell extravasation, tumor penetration, tumor killing, T cell dependence, HIV

suppression, and compatibility with anti-HIV medication were assessed. mNK cells can be new players in absence of CD4 help that can contribute to clear head and neck cancer cells and can be soldiers for HIV cure strategy.

Comments:

mNK proportion of these cells among NK should be provided as well as tissue distribution. Patients living with HIV+ cancer have been excluded from immunotherapy for two main reasons¹⁶: 1) concerns about undesired interactions between ART drugs and the immunotherapy treatment; 2) risk of accelerating HIV-1 progression due to the checkpoint inhibitors, cytokines, or immune cell injections used in immunotherapy².

Such statement was true years ago not now, immunotherapies are available now, should be updated

This paper is a lab paper too many unsubstantiated information on HIV therapy or cancer treatment are presented

This paper should be limited to the lab work and rewritten as such as global consideration for HIV are outdated or incorrect.

Reviewer #3 (Remarks to the Author):

This manuscript describes an elegant system to assess the effectiveness of memory-like natural killer cells (mNK) against head and neck squamous cell carcinoma (NHSCC) in people with HIV (PLH). They use a 3D culture system to show that mNK cells can kill NHSCC cells if they are exposed to HIV-infected T cells. The paper is of substantial interest in advancing the field of immunotherapy of HIV cancers. However, I had several comments and concerns.

1. The focus of their manuscript is to show that HIV patients can mediate killing of HNSCC by mNK cells. Ideally, they would show that mNK cells developed from PLH work in this system, but they do not show this. In fact, they do not really look at the effects of HIV infection on mNK activity against HNSCC in their in vitro system.

2. The author state a number of times that this is one of the first studies examining immunotherapy of tumors in HIV patients. While such patients have been excluded from many trials, there have in fact been some studies of checkpoint inhibitors in HIV patients, and the NCI and FDA both now have guidelines to include such patients. The authors should more accurately reference this literature and clinical trial environment.

3. The authors use this as a model to study mNK activity against HIV+ NHSCC. They study "patient derived HNSCC cells; however, it isn't clear if the tumors studied were in fact from HIV+ patient. Moreover, many HNSCC tumors in HIV patients are caused by and express human papillomavirus (HPV); were these HPV+ tumors? Also, they do not give detailed information about how these cells were obtained; this information should be provided.

4. There appear to be a number of small mistakes in the presentation of the data:

a. Figure 1, F and G seem to be mislabeled.

b. Fig 2D is not really explained in the legend.

c. Fig 3 H. It is unclear what the different panels are or what the white arrows are pointing to.

5. In general, the figure legends (or text) should provide more information of the specifics of how the experiments were conducted. For example, the figures generally do not show the number of replicates, and the statistical section just says that all experiments were done at least 3 times. In Fig 4A, for example, they say that a change from 85 to 93% is statistically significant, but one would like to have a sense of how many replicates this represented; such a change is usually no significant unless there are multiple replicates.

REVIEWER COMMENTS

Reviewer #1 (Remarks to the Author):

The manuscript by Ayuso et al describes studies involving use of an in vitro microphysiological system (MPS) to evaluate the functionality of human cytokine-induced memory-like NK cells against HNSCC tumor cells in the presence or absence of human HIV-infected CD4+ T cells. This MPS platform represents a novel and potentially powerful tool to study NK cell-mediated defense against tumors in settings of HIV infection, which is currently limited by the lack of a small animal models of HIV and HIV-related cancers that recapitulate aspects of human disease. In the MPS system, NK cells (and HIV-infected CD4+ T cells) traffic out of endothelium-lined channels and into a tumor-containing tissue-like microenvironment where the NK cells mediate cytolysis of the tumor cells. Overall, this is an exciting study that is expected to appeal to a broad audience within the immunology field, in part due to the potential utility of the platform in addressing other open questions in the field in addition to those evaluated in this particular study.

We thank the reviewer for their encouraging words and constructive feedback. We have modified the manuscript following their comments and we believe these changes have helped us to strengthen the manuscript.

Additionally, this manuscript was submitted initially to Nature Letters, which has a very distinctive format: 2,000 words max in a main section (i.e., no introduction, results, or discussion sections). Thus, we have also modified the paper structure to include individual sections such as introduction, results, and discussion. Since the 2,000-word limit is not applicable for papers in Nature Communications, we were also able to expand the level of detail following the reviewers' suggestions.

Nevertheless, several important concerns were noted. Most importantly, it is difficult to discern whether many of the key findings are reproducible within and across donors because little or no information is provided regarding the number of donors evaluated or the number of experimental replicates for each assay. Inter-donor variation in outcomes is a concern and high variation could seriously limit the impact of the findings. For all experiments, the number of technical replicates and intra- and inter-donor replicates should be clearly stated within each figure legend and results using cells from multiple donors should be shown so that inter-donor variability can be assessed.

The reviewer is right that inter-donor variation in adoptive cell therapy is a relevant point. Although the main point of this work was not to evaluate inter-donor variation across a large patient cohort, we suggest the inclusion of a new figure (new Supporting Figure 3, included below for reference), where we present data showing ML-NK cell cytotoxicity among multiple donors. We also discussed these ideas in the manuscript:

“Interestingly, ML-NK, naïve NK, and NK-92 cells exhibited a similar cytotoxicity profile against HSNCC samples (i.e., sample #CK7521 was the most vulnerable whereas sample #CK5566 remained the most resistant). Additionally, ML-NK cells obtained from multiple donors exhibited a consistent cytotoxicity pattern against HNSCC samples (i.e., sample #CK7521 was the most vulnerable, #CK5566 was the most

resistant). However, ML-NK cells from multiple donors differ in their cytotoxic potential, highlighting the role of donor heterogeneity and the importance of biomarkers that allow us to successfully predict ML-NK cell efficacy (Supporting Figure 3). In this context, multiple studies have explored the role of activator (e.g., NKG2D) and inhibitory NK cell receptors (e.g., NKG2A) as well as immune exhaustion markers as predictive tools.”

” In this study, we analyzed a small number of NK cell donors. Although we tested a limited pool of donors, our results showed that ML-NK cells obtained from different donors exhibited significant differences in cytotoxic potential (Supporting Figure 3). Factors such as the repertoire and level of expression of NKG2 receptors expressed in ML-NK cells could have a deep impact on cytotoxicity against tumor cells. Further studies may also analyze whether patient-variability also leads to differences in ML-NK cell resistance to hypoxic and nutrient-depleted environments. We believe that future studies evaluating ML-NK cell inter-donor variation across dozens, or hundreds of donors could help to define a molecular signature to optimize treatment response in a patient specific manner.”

Supplementary Figure 3. Evaluation of ML-NK cell cytotoxicity against patient-derived HNSCC samples. Primary NK cells were isolated from multiple donors and differentiated into ML-NK cells by cytokine stimulation. After 7 days, they were co-cultured with HNSCC cell lines (i.e., SCC-6) and HNSCC patient-derived samples for another 24 hours to evaluate their cytotoxic potential. *, **, and **** indicate p-value < 0.05, 0.01, 0.001 respectively. Brackets indicate the comparison of all the control samples with the ML-NK samples done by a nested t-test.

Similarly, many critical experimental details are omitted in the manuscript, which makes it broadly difficult to assess the rigor of several experiments. These include the following:

- For all image-based data, a clear and detailed description of the methods used to analyze the images, including the number of images/field/MPS replicates evaluated for summary data, must be included.

We agree with the reviewer that adding this data will be valuable for the reader. Thus, we have expanded the image analysis section in the materials and method section and modified all the figure legends including image-based data to include a more detailed description. We have also included

additional details in the figure legends to describe the experimental design. See below an example from Figure 1E (only new text is included):

Fig 1 E. Protocol followed to study NK cell immunotherapy response in HIV⁺ HNSCC [...]. Fluorescence images show a representative experiment. The experiment was repeated three independent times, acquiring two images per experiment. Images from the same experiment were averaged and used as an independent replicate. Graphs show average of these three independent replicates \pm standard deviation. P-value was set to 0.05.

See below the text added to the main manuscript regarding image/field of view analysis:

“For NK-92, naïve NK, and ML-NK cell experiments analyzing NK cell cytotoxicity, tumor penetration, extravasation, and HIV detection, five technical replicates were acquired per condition, and experiments with primary cells were repeated at least with three different donors. We used a Leica microscope equipped with an automated stage and tiles-scan capabilities. This allowed us to acquired multiple 10X images and merge them together to generate one image that covered a large area while preserving high image resolution at single cell level. To show inter-donor variability, data obtained from technical replicates were averaged to obtain the value of a given donor or condition.”

- It is unclear what type of transcriptomic analyses (e.g. Fig. 2, Suppl Fig. 5) were performed for this study. The method (i.e. RNAseq) should be stated, the methods used for data analyses must be described, and the raw data should be made available.

The reviewer is right that the section describing the transcriptomic analysis was missing. We have added a description explaining the experimental protocol as well as the method used to generate the data. See modified text below:

“Gene expression analysis by RT-qPCR array

To study how NK-92, naïve NK, and ML-NK cells adapted gene expression to the different conditions evaluated (e.g., cultured in the presence of CD4 T cells, difference between naïve vs ML-NK cell), we used RT-qPCR arrays from Qiagen (RT2 Profiler). These arrays allow the simultaneous analysis of up to 384 genes related with immune activation/exhaustion. After isolating the desired cell type (e.g., NL-NK cells), cells were lysed, and mRNA was captured using the Dynabead mRNA DIRECT Purification Kit (Thermo Fisher Scientific, 61011). Isolated mRNA was quantified using a Qubit fluorometer (Thermo Fisher Scientific) and the Qubit RNA BR Assay Kit (Q10210, Thermo Fisher Scientific). mRNA was reverse-transcribed to complementary DNA (cDNA) using the RT2 PreAMP cDNA Synthesis Kit (Qiagen, 330451). cDNA quantity was analyzed by RT-qPCR using the “Cancer Inflammation and Immunity Crosstalk” Qiagen RT2 profiler panel (Qiagen, PAHS-181Z), and data were analyzed using the Qiagen online software (<https://geneglobe.qiagen.com/us/analyze/>) (fig. S5). These panels include several controls that are also analyzed by the Qiagen software, ensuring sample quality (e.g., positive control, retrotranscription control, contamination control) and only samples that passed all these tests were used in the analysis. Clustergrams and volcano plots were generated using Qiagen’s software, bar graphs were generated in GraphPad 9.0, and Gene Set Enrichment Analysis (GSEA) graphs were generated through (<https://www.gsea-msigdb.org/gsea/index.jsp>)”.

We also make the raw data available for potential readers.

- Data in Figs. 2B, D, and E would be best conveyed as a heat map showing relative expression with individual up- and down-regulated genes identified. Also, it is unclear what the yellow color in Fig 2D signifies.

Following the reviewer’s suggestion, we have modified Figure 2 by removing panel D and adding instead a heat-map showing individual up- and down-regulated genes. As the reviewer suggested, this panel will allow readers to see up and down-regulated genes individually in a more direct fashion. See new Figure 2 below (we are only showing a fragment of figure 2 due to the large size of Figure 2, full figure can be found in the main manuscript):

We have modified the figure legend to reflect these changes.

“D) Clustergram showing differentially expressed genes in ML-NK cells compared with naïve NK cells.”

- Page 6, please state the actual range of concentrations found in plasma from HIV patients so that concordance with the assay conditions can be better evaluated.

We agree with the reviewer’s comment, providing such comparison will allow the reader to put our results into perspective. When we selected the range concentration for the several ART drugs analyzed, we tried to ensure they were within the physiological range. The ART range used in our experiments went from 0.01nM to 1mM, *in vivo* studies described plasma concentrations of 10.8 μM , 1.6 μM , and 1.1 μM for Doravirine, Abacavir, and Atazanavir respectively after 5 hours post oral intake. Thus, we believe our range includes physiologically relevant concentrations.

We have included a statement in the manuscript describing the drug concentrations found in plasma in people living with HIV:

“We analyzed a range of concentrations consistent with those found in plasma from patients with HIV (Max concentration found in plasma: Doravirine= 10.8 μM^1 ; Abacavir 1.6 μM^2 , Atazanavir 1.1 μM^3 within 1-5 h post oral intake)”

- For all subpanels in Fig. 3 and 4: please clarify/state in legend (1) which culture system was used for these studies - i.e. MPS shown in Fig.1 or another system; (2) whether all NK plus T cell co-cultures utilized autologous cells; and (3) whether NK-92 or primary NK cells were used.

As the reviewer pointed out, those figures include multiple experimental set ups, which may lead to confusion in the audience. Thus, following the reviewer’s suggestion, we have added a statement in the figure legend stating the experimental configuration used in all panels shown in Fig.3 and 4 (see example below):

“Figure 4[...] Experiments shown in panels C-L were performed in the MPS.”

In figures 3 and 4, we used allogenic ML-NK and T cells, since we were interested in allogenic ML-NK cell infusion in HIV⁺ HNSCC patients. We have added a statement clarifying that detail in figures using ML-NK cells (see example below):

“Allogenic primary ML-NK and CD4 T cells were used in these experiments”

In Fig. 3 and 4, we exclusively used primary ML-NK cells in these experiments since we thought they had more relevance than NK-92s for the questions posed. Additionally, we conducted similar experiments using NK-92 cells and included those findings in supplementary figures (e.g., supplementary figure 2, 4, 10, and 11). We have ensured all figure legends stated the type of NK and T cell used (i.e., naïve vs ML-NK vs cell line, autologous vs allogenic).

- Fig. 3H: Please indicate what does each column represents and what the arrows in images represent.

Following the reviewer’s suggestions, we have added additional information to Figure 3H legend and we have rearranged panel H to make it clearer (see new Figure below). Left panel shows HIV-infected T cells and ML-NK cells right after seeding them in co-culture (i.e., 1 hour post co-culture with ML-NK cells). Right panel shows the same culture after 24 hours in co-culture, allowing ML-NK cells to cluster around HIV-infected T cells. The arrows show the location of HIV-infected CD4 T cells (labelled in green due to the HIV fluorescent reporter). We have also included a magnification of these images to allow the reader to clearly observe ML-NK cell clustering around infected T cells. We added these explanations to the figure legend as well (see below panel H and legend for reference):

“G-I) ML-NK cells recognize HIV infection in CD4 T cells. Primary CD4 T cells were infected with a single-round (*Env-minus*) HIV dual-color fluorescent reporter and cultured for 24 hours to allow the virus to reach the late stage of the viral cycle (detected by GFP expression on CD4 T cells). Next, HIV-infected CD4 T cells were co-cultured with ML-NK cells (in cyan). Microscopy images demonstrate that ML-NK cells clustered around the HIV-infected CD4 T cells, whereas most of the non-infected T cells remained as single cells. Arrows indicate the location of HIV-infected CD4 T cells (labelled in green) and show that most T cells had ML-NK cells in contact. Graphs in panel I shown the percentage of T cells with (in white) and without (in grey) ML-NK cells attached I the absence (left graph) presence of HIV-infection (right panel). Experiments shown in panels G-I were performed in 96 well-plates.

“When we co-cultured ML-NK cells with autologous HIV-infected CD4⁺ T cells (i.e., CD4 T cells and ML-NK cells came from the same patient), we observed that ML-NK cells clustered around the infected T cells, whereas non-infected T cells remained as isolated cells (Figure 3H-I). This result suggested that ML-NK cells were capable of detecting the presence of the virus inside of the T cells (Figure 3H, bottom inserts show a magnification of ML-NK cell clustering around HIV-infected T cells). We also repeated the experiment combining allogenic ML-NK cells with HIV-infected T cells (i.e., obtained from different patients). The results demonstrated that allogenic ML-NK cells recognized HIV-infected T cells and left non-infected T cells as single cells (Supporting Figure 6).”

- Fig. 3I: Please clarify the figure legends and X axis labels, which are small and confusing.

We agree that there were axis marks in the X axis that were misleading for the audience. We have removed them. New graphs in Fig. 3I show the percentage of T cells with/without ML-NK cells attached before and after HIV infection. We have enlarged the graph legend that is located at the top of these graphs and also added this information into the legend:

Graphs in panel I show the percentage of T cells with (in white) and without (in grey) ML-NK cells attached in the absence (left graph) presence of HIV-infection (right panel).

It is also possible that during the file conversion process some of the smaller details in these graphs lost resolution. We will provide higher resolution figures for the final manuscript.

- Supporting Fig. 3A: Primary data should be shown, not “+, ++, etc.” designation, which is not quantitative.

We have removed old panel 3A and combined the gene expression data with other data requested by the reviewer (i.e., cytotoxicity inter-donor variability). Please, see new supplementary Figure 3 below:

“Supplementary Figure 3. Evaluation of ML-NK cell cytotoxicity against patient-derived HNSCC samples. Primary NK cells were isolated from multiple donors and differentiated into ML-NK cells by cytokine stimulation. After 7 days, they were co-cultured with HNSCC cell lines (i.e., SCC-6) and HNSCC patient derived samples for another 24 hours to evaluate their cytotoxic potential. *, **, and **** indicate p-value < 0.05, 0.01, 0.001 respectively. Brackets indicate the comparison of all the control samples with the ML-NK samples done by a nested t-test. C) We analyzed the expression of multiple genes associated with NK cell sensitivity and activation. The normalized fold change expression shown in the graph was calculated as follows: first we calculated the fold change of each individual gene with respect the average of five housekeeping genes, next we normalized that fold change with respect the fold changed obtained in CK7521 cells. Thus, this provides the fold change of each gene in CK8257 and CK5566 cells normalized to the value obtained in CK7521. *, **, *** denote p-value<0.05, <0.01, <0.005. ”

- Supporting Fig. 3B: Describe how fold change was calculated

We have included a statement explaining the process in the figure legend. See below:

C) We analyzed the expression of multiple genes associated with NK cell sensitivity and activation. The normalized fold change expression shown in the graph was calculated as follows: first we calculated the fold change of each individual gene with respect to the average of five housekeeping genes for a given HNSCC sample, next we normalized that fold change with respect to the fold change obtained in CK7521. Thus, this provides the fold change of each gene in CK8257 and CK5566 cells normalized to the value obtained in CK7521.

- Differences in outcomes between autologous and allogeneic NK and CD4 T cell co-culture systems should be evaluated and quantified.

NK cells may offer an interesting alternative to T cells for allogeneic transplant. Thus, in this manuscript, we wanted to explore the potential of allogeneic ML-NK cells to treat HIV⁺ HNSCC. All the experiments except one were performed with allogeneic ML-NK cells and T cells. The only experiment where we explored autologous vs allogeneic ML-NK and T cells is the HIV-infection recognition. Supporting Figure 6 showed that allogeneic ML-NK cells were able to detect HIV-infection in T cells, but we wondered whether the allogeneic nature of these cells could be causing this recognition/clustering around HIV-infected T cells. Thus, in Figure 3G-I, we used autologous ML-NK and T cells. The results demonstrated that autologous ML-NK cells were also able to recognize HIV infection within T cells. We agree that depending on multiple molecular factors (e.g., NK cell expression of NKG2 receptors vs T cell expression of HLA molecules), allogeneic ML-NK cells may exhibit a broad range of HIV-infection recognition. We have modified the text to discuss all these concepts:

“One of the limitations of our study is the low number of NK cell donors tested. Inter-donor variability in immunotherapy outcome is a relevant topic for both clinicians and researchers. Although we tested a limited pool of donors, our results showed that ML-NK cells obtained from different donors exhibited significant differences in cytotoxic potential (Supporting Figure 3). Factors such as the repertoire of NKG2 receptors expressed in ML-NK cells could have a deep impact on cytotoxicity against tumor cells. Further studies may also analyze whether patient-variability also leads to differences in ML-NK cell resistance to hypoxic and nutrient-depleted environments. We believe that future studies evaluating ML-NK cell inter-donor variation across dozens, or hundreds of donors could help to define a molecular signature to optimize treatment response in a patient specific manner.”

Other minor issues:

- The abbreviation for “memory-like NK cells” should be “ML-NK” cells, not “mNK” cells. The latter is often used as an abbreviation for mature NK cells.

We thank the reviewer for this point since that nomenclature could have led to confusion. We have changed the abbreviation to “ML-NK cells”

- Define PDMS at first instance.

Fixed, we defined PDMS the first time it was mentioned.

- When discussing memory-like NK responses in the text, the authors should be careful to discriminate between antigen-specific/-dependent recall responses by NK cells, which are broadly considered bona fide memory responses, and cytokine-induced memory-like responses that are antigen non-specific.

This distinction adds a relevant context for the manuscript. Thus, we have added a paragraph discussing these differences in NK cell memory/memory-like response in the discussion:

“As described, previous studies have demonstrated that antigen exposure leads to the generation of recall responses by NK cells that are considered a bona fide memory response. In this study, all the ML-NK cells used were generated by exposing naïve NK cells to a specific cocktail of cytokines, leading to the generation of an antigen-agnostic, memory-like phenotype.”

Following Reviewer #2’s first comment, we have significantly expanded the introduction to discuss in more details ML-NK cells. Please, see his comment below for additional detail.

- A figure should be referenced after this sentence: “Additionally, mNK cells showed significantly lower cytokine-dependence (e.g., IL-12) to exert their cytotoxic function”.

We have cited the figure panel showing higher killing in ML-NK cells compared with primary NK cells (Figure 2F).

- Fig. 3A and Supporting Fig 4. Please replace "vs" with "plus" or "+/-", since the comparison is NKs with or without ART, not mKs vs ART.

Fixed

- Page 9: Supporting Figures numbers are mislabeled in the text

We have reviewed the numbers in supporting figures. Please, let us know if the reviewer still identifies any potential error in figure numbering.

- Fig. 4A: Please label T cells in diagram

Fixed

- Please correct typos throughout the manuscript.

We have reviewed the manuscript and addressed all the typos that we found

Reviewer #2 (Remarks to the Author):

Human-derived microphysiological models reveal the promise of memory-like NK cell immunotherapy for patients with HIV+ cancer

Metabolism

Can we share the cancer immunotherapy revolution with HIV patients having cancer. Investigators leveraged human-derived microphysiological models to reverse-engineer the HIV-immune system interaction and evaluated the potential of memory-like natural killer (mNK) cells for HIV head and neck cancer, a non-AIDS cancer related to tobacco or HPV.

mNK cells are different as they have strong cytotoxicity and activation, and live long-term. Different mNK cell response in human-derived bioengineered HIV+ environments, including mNK cell extravasation, tumor penetration, tumor killing, T cell dependence, HIV suppression, and compatibility with anti-HIV medication were assessed.

mNK cells can be new players in absence of CD4 help that can contribute to clear head and neck cancer cells and can be soldiers for HIV cure strategy.

Comments:

mNK proportion of these cells among NK should be provided as well as tissue distribution.

Following the reviewer's suggestion, we have added additional information discussing ML-NK cell expansion and tissue distribution in the introduction. Please, see new text for reference:

“Traditionally, NK cells have been considered part of the innate system due to a variety of reasons including their rapid activation in the absence of prior antigen exposure and the lack of somatically rearranged receptors, as observed in T cells. However, in the last years, mounting scientific evidence has challenged such classification, showing that NK cells share several critical features typically ascribed to the adaptive response, such as antigen-specific, clonal-like expansion; as well as the development of a persistent memory-like cell pool that exhibits strong recall responses after repeated antigen exposure. Early studies in mouse models revealed that after exposure to mouse cytomegalovirus (MCMV), ML-NK cells persisted in the mouse's blood for weeks and even months. Subsequent human studies revealed that exposure to CMV leads to NKG2C⁺ ML-NK cell expansion, which are still detected in the patients' blood after a year even in the absence of detectable CMV viremia. Subsequent studies demonstrated NK cells develop a ML phenotype in response to a variety of bacteria (e.g., Mycobacterium) and viruses, including influenza, HIV, CMV, Epstein-Barr, or Herpes simplex virus. However, our knowledge about the mechanisms controlling ML-NK cell tissue distribution remains limited. Hapten-derived ML-NK cells concentrate in the liver, while influenza-specific ML NK cells are present in liver and lung. Finally,

cytokine-induced ML-NK cells have been detected systematically, highlighting their potential to treat hematological cancers. However, evaluating ML-NK cell potential for HIV+ solid tumors remains challenging due to the complexity of the tumor microenvironment and HIV-1 infection.”

Patients living with HIV+ cancer have been excluded from immunotherapy for two main reasons¹⁶: 1) concerns about undesired interactions between ART drugs and the immunotherapy treatment; 2) risk of accelerating HIV-1 progression due to the checkpoint inhibitors, cytokines, or immune cell injections used in immunotherapy².

Such statement was true years ago not now, immunotherapies are available now, should be updated

Following the reviewer’s comment, we have removed that statement. We have modified the text to state that PLWH can have access to immunotherapies with immune checkpoint inhibitors. We are also adding some information showing the exclusion criteria used in clinical trials exploring adoptive cell therapy (i.e., injection of immune cells):

“Immune checkpoint inhibitor guidelines are rapidly evolving for immunocompromised patients, making these treatments accessible for this population. Multiple reviews have found that immune checkpoint inhibitors in HIV+ cancer patients are well tolerated, showing clinically relevant antitumor activity and improving patient outcomes. Additionally, immune checkpoint inhibitors in HIV+ cancer patients are not associated with adverse changes in HIV viremia or CD4 T cell count. However, clinical trials exploring adoptive cell immunotherapy commonly still exclude immunocompromised individuals (NCT03296137, NCT04729543, NCT05451784, NCT03068819, NCT02782546). This situation is especially relevant for the almost forty million people living with HIV worldwide.”

This paper is a lab paper too many unsubstantiated information on HIV therapy or cancer treatment are presented. This paper should be limited to the lab work and rewritten as such as global consideration for HIV are outdated or incorrect.

Following the reviewer’s suggestion, we have modified multiple sections of the manuscript to focus in the *in vitro* findings, minimizing references to global considerations. We have included several statements highlighting the limitations of our study (see below some examples):

“This study exclusively relies on in vitro systems, and we have limited our study to a few critical parameters involved in HIV-cancer-immune system interactions. Thus, additional studies combining MPS with humanized animal models would enable us to create providing an improved pipeline to test new treatments (e.g., ML-NK cells) across different types of immunosuppression (e.g., organ transplant patients) to provide a more robust menu of therapeutic options for challenging patients.”

“One of the limitations of our study is the absence of a direct comparison between HPV+ and HPV- HNSCC since all our samples were negative for HPV, HHV, Epstein-Barr, or HIV. HPV infection plays a critical role in HNSCC progression and treatment response”.

“In this study, we analyzed a small number of NK cell donors. [...] Factors such as the repertoire and level of expression of NKG2 receptors expressed in ML-NK cells could have a deep impact on cytotoxicity against tumor cells. Further studies may also analyze whether patient-variability also leads to differences in ML-NK cell resistance to hypoxic and nutrient-depleted environments.”

Additionally, this manuscript was submitted initially to Nature Letters, which has a very broad scope and distinctive format: 2,000 words max in a main section (i.e., no introduction, results, or discussion sections). Thus, we have also modified the paper structure to include individual sections such as introduction, results, and discussion. These changes allowed us to focus on the details found in our in vitro experiments and we believe this new version provides a more balanced version.

If the reviewer still identifies specific statements that they believe we should modify, we will be happy to address the issue.

Reviewer #3 (Remarks to the Author):

This manuscript describes an elegant system to assess the effectiveness of memory-like natural killer cells (mNK) against head and neck squamous cell carcinoma (HNSCC) in people with HIV (PLH). They use a 3D culture system to show that mNK cells can kill HNSCC cells if they are exposed to HIV-infected T cells. The paper is of substantial interest in advancing the field of immunotherapy of HIV cancers.

We thank the reviewer for their encouraging words and constructive feedback. We have modified the manuscript following their comments and we believe these changes have helped us to strengthen the manuscript.

Additionally, this manuscript was submitted initially to Nature Letters, which has a very distinctive format: 2,000 words max in a main section (i.e., no introduction, results, or discussion sections). Thus, we have also modified the paper structure to include individual sections such as introduction, results, and discussion. Since the 2,000 word limit is not applicable for Nature Communications, we were also able to expand the level of detail following the reviewers' suggestions.

However, I had several comments and concerns.

1. The focus of their manuscript is to show that HIV patients can mediate killing of HNSCC by mNK cells. Ideally, they would show that mNK cells developed from PLH work in this system, but they do not show this. In fact, they do not really look at the effects of HIV infection on mNK activity against HNSCC in their in vitro system.

The reviewer is discussing a relevant topic and we agree that studying the capacity of the patient's ML-NK cells to kill HIV-infected cells could lead to new breakthroughs. However, the intent of our manuscript was not to study the HIV⁺ patient's own NK cells but to focus on the potential of ML-NK cells for allogenic transplant (i.e., from healthy donors to HIV⁺ cancer patients).

However, we believe the reviewer's suggestions also brings value to the manuscript, thus we have expanded the introduction and discussion to: 1) highlight the role of autologous ML-NK cells in viral progression; and 2) emphasize the fact that the scope of this study were allogenic ML-NK cells:

“Early studies in mouse models revealed that after exposure to mouse cytomegalovirus (MCMV), ML-NK cells persisted in the mouse’s blood for weeks and even months⁸. Similarly, exposure to human CMV leads to NKG2C⁺ ML-NK cell expansion in humans, which are still detected in the patients’ blood after a year even in the absence of detectable CMV viremia⁹. Subsequent studies demonstrated NK cells develop a ML phenotype in response to a variety of bacteria (e.g., Mycobacterium) and viruses, including influenza, HIV, CMV, Epstein-Barr, or Herpes simplex virus⁹. Other reports have shown that CMV infection can lead to the generation of a NKG2C⁺ ML-NK cell subset that reduces HIV viremia during primary infection, highlighting the critical role of ML-NK cells in HIV progression ¹⁰.”

“[...] Altogether, these studies highlighted ML-NK cells play a critical role in HIV and tumor progression, suggesting therapies based on injections of allogenic ML-NK cells could provide a versatile tool for HIV⁺ cancer patients.”

2. The author state a number of times that this is one of the first studies examining immunotherapy of tumors in HIV patients. While such patients have been excluded from many trials, there have in fact been some studies of checkpoint inhibitors in HIV patients, and the NCI and FDA both now have guidelines to include such patients. The authors should more accurately reference this literature and clinical trial environment.

The reviewer is right that treatment guidelines for immune checkpoint inhibitors consider the inclusion of HIV⁺ patients. Cancer immunotherapy is a broad term that encompasses a variety of treatments such as checkpoint inhibitors or adoptive cell therapy. The scope of our paper was to focus on cell therapy, more specifically on allogenic transplant of ML-NK cells. Thus, following the reviewer's suggestion, we have modified the text throughout the manuscript to provide a more holistic discussion. We have: 1) stated that new guidelines for checkpoint inhibitors include HIV⁺ cancer patients (we have included additional references); and 2) cited multiple clinical trials using adoptive T, NK, or NK cell therapy where HIV is an exclusion factor.

“Immune checkpoint inhibitor guidelines are rapidly evolving for immunocompromised patients, making these treatments accessible for this population. Multiple reviews have found that immune checkpoint inhibitors in HIV⁺ cancer patients are well tolerated, showing clinically relevant antitumor activity, and improving patient outcomes. Additionally, immune checkpoint inhibitors in HIV⁺ cancer patients are not

associated with adverse changes in HIV viremia or CD4 T cell count. However, clinical trials exploring adoptive cell immunotherapy commonly still exclude immunocompromised individuals (NCT03296137, NCT04729543, NCT05451784, NCT03068819, NCT02782546). This situation is especially relevant for the almost forty million people living with HIV worldwide.”

3. The authors use this as a model to study mNK activity against HIV+ HNSCC. They study “patient derived HNSCC cells; however, it isn’t clear if the tumors studied were in fact from HIV+ patient. Moreover, many HNSCC tumors in HIV patients are caused by and express human papillomavirus (HPV); were these HPV+ tumors? Also, they do not give detailed information about how these cells were obtained; this information should be provided.

Tumors isolated from HIV⁺ and HIV⁻ patients will likely present multiple additional differences besides HIV status such as HPV infection, EGFR, PD-1, CTLA-4, and expression of other molecular factors, which in turn would make more challenging any interpretation regarding ML-NK cell efficacy. Finding two patients with exactly the same molecular and medical makeup except for their HIV infection status is complicated. Thus, our goal was to generate a microphysiological system that included a HNSCC tumor where we could control and manipulate the presence of HIV and HIV-infected/non-infected CD4 T cells.

Regarding sample isolation, we see value in the reviewer’s comment, and we have modified the text to include a detailed protocol describing cell isolation as well as the HIV (and other virus) status:

“HNSCC surgical samples were collected from the UW-biobank and cut into <1mm pieces using a surgical blade. Tumor fragments were digested in RPMI-1640 supplemented with collagenase 1 (8mg/mL), collagenase 4 (4mg/mL), hyaluronidase (4 mg/mL), and DNase I 1mg/mL (all obtained from Worthington Biochemical Corporation) for 4 hours at 37 °C in continuous agitation. Next, the tissue/cell suspension was centrifuged at 400g for 5 min, the pellet was resuspended in PBS and filtered through a sterile 100-µm filter to remove ECM fragments. Finally, the filtered cell suspension was cultured in Advanced DMEM/F-12 (Invitrogen), supplemented with 5% FBS, 0.4 ng/mL hydrocortisone (Sigma), 10 ng/mL EGF (Invitrogen), 24 µg/mL adenine in 1mM HCl, Penicillin/Streptomycin, 2 mM L-Glutamine, and 10 µM Y-27632 dihydrochloride. NK-92 cells were cultured in X-VIVO 15 (Lonza) supplemented with 20% FBS (VWR) and 100 U/mL IL-2. J-lat cells were cultured in RPMI Medium 1640 (1X) + L-Glutamine supplemented with 10% FBS (VWR) and 1% Penicillin/Streptomycin and HIV-1 expression was induced by exposing J-lat cells to 10 ng/mL TNF-α. Samples included in this study were negative for HIV, HPV, HHV, EBV”.

We have also added a discussion on the role of HPV in NK cell immunosurveillance during HNSCC:

“One of the limitations of our study is the absence of a direct comparison between HPV⁺ and HPV⁻ HNSCC since all our samples were negative for HPV, HHV, Epstein-Barr, and HIV. HPV infection plays a critical role in HNSCC progression and treatment response. HPV⁺ HNSCC shows higher infiltration of cytotoxic NK cells (CD56^{dim}). Additionally, HPV⁺ HNSCC exhibits lower HLA class I expression, which include ligands for several inhibitory NK cell receptors (e.g., NKG2A). In this context, low HLA class I expression is associated

with better prognosis for HPV⁺ HNSCC, but worse outcome in HPV⁻ tumors; suggesting HPV infection contributes to the NK cell response against HPV⁺ HNSCC.”

4. There appear to be a number of small mistakes in the presentation of the data:

a. Figure 1, F and G seem to be mislabeled.

We thank the reviewer for bringing up these errors. We have addressed them.

b. Fig 2D is not really explained in the legend.

We have expanded the details provided in the legend to include a description of panel 2D. Please note that following the suggestions proposed by reviewer #1, new panel 2D has been substituted by a clustergram with genes upregulated and downregulated in ML-NK cells compared to naïve NK cells. See modified panel below:

“D) Clustergram showing differentially expressed genes in ML-NK cells compared with naïve NK cells.”

c. Fig 3 H. It is unclear what the different panels are or what the white arrows are pointing to.

Following the reviewer’s suggestion, we have added additional information to Figure 3H and we have rearranged panel H to make it easier to understand for the audience (see new Figure below). Fig 3H, left panel shows HIV-infected T cells and ML-NK cells right after seeding them in co-culture (i.e., 1 hour post co-culture). Right panels show the same culture after 24 hours in co-culture, providing ML-NK cells enough time to cluster around HIV-infected T cells. The arrows show the location of HIV-infected CD4 T cells (labelled in green due to the HIV fluorescent reporter). We have also included a magnification of these images to allow the reader to clearly observe ML-NK cell clustering around infected T cells. We added these explanations to the figure legend as well (see below new text, new panel H, and modified

legend for reference):

“When we co-cultured ML-NK cells with autologous HIV-infected CD4⁺ T cells (i.e., CD4 T cells and ML-NK cells came from the same patient), we observed that ML-NK cells clustered around the HIV-infected T cells, whereas non-infected T cells remained as isolated cells (Figure 3H-I). This result suggested that ML-NK cells are capable of detecting the presence of the virus inside of the T cells (Figure 3H, bottom inserts show a magnification of ML-NK cell clustering around HIV-infected T cells). We also repeated the experiment culturing allogenic ML-NK cells with HIV-infected T cells (i.e., obtained from different patients). The results demonstrated that allogenic ML-NK cells also recognized HIV-infected T cells and left non-infected T cells untouched (Supporting Figure 6).”

“G-I) ML-NK cells recognize HIV infection in CD4 T cells. Primary CD4 T cells were infected with a single-round (Env-minus) HIV dual-color fluorescent reporter and cultured for 24 hours to allow the virus to reach the late stage of the viral cycle (detected by GFP expression on CD4 T cells). Next, HIV-infected CD4 T cells were co-cultured with ML-NK cells (in cyan). Microscopy images demonstrate that ML-NK cells clustered around the HIV-infected CD4 T cells, whereas most of the non-infected T cells remained as single cells. Arrows indicate the location of HIV-infected CD4 T cells (labelled in green), showing most T cells had ML-NK cells in contact. Graphs in panel I shown the percentage of T cells with (in white) and without (in grey) ML-NK cells attached I the absence (left graph) presence of HIV-infection (right panel). Experiments shown in panels G-I were performed in 96 well-plates.

5. In general, the figure legends (or text) should provide more information of he specifics of how the experiments were conducted. For example, the figures generally do not show the number of replicates, and the statistical section just says that all experiments were done at least 3 times. In Fig 4A, for example, they say that a change from 85 to 93% is statistically significant, but one would like to have a sense of how many replicates this represented; such a change is usually no significant unless there are multiple replicates.

Following the reviewer’s suggestions, we have added additional details to the figure legends, including

the number of individual donors, the number of technical replicates from each donor, and the data showed in quantitative graphs. Please, see modified legend from Figure 1F as an example:

“E) HIV-1 infection of CD4 T cells was monitored with genetically modified single-round HIV-1 viruses encoding dual fluorescent reporters. Primary CD4 T cells infected with HIV-1 express the fluorescent protein mCherry and YFP during early and late-stage infection. Fluorescence images show a representative result. The experiment was repeated using CD4 T cells from three different blood donors, acquiring five images per experiment. Images from the same experiment were averaged and used as an independent replicate. Graphs show average of these three independent replicates \pm standard deviation. P-value was set to 0.05”

Regarding Figure 4A, the small error bars and statistical significance are partly due to the experimental design: to analyze NK cell cytotoxicity in 2D (e.g., Fig 4A) we used a Leica microscope equipped with automated stage and tile-scan capabilities, which allowed us to acquire large field of views that encompassed most of the 96 well-plate using a 10X objective (which allowed us to preserve high resolution at single cell level as shown in the images). Additionally, each experiment using cells from blood donors, was designed to include 5 technical replicates (i.e., 5 wells of the 96 well-plate), which combined with the large field of view scanned, helped us to minimize experimental variability. We have included these details in the materials and methods section:

“For NK-92, naïve NK, and ML-NK cell experiments analyzing NK cell cytotoxicity, tumor penetration, extravasation, and HIV detection, five technical replicates were acquired per condition, and experiments with primary cells were repeated at least with three different donors. We used a Leica microscope equipped with an automated stage and tile-scan capabilities. This allowed us to acquire multiple 10X images and merge them together to generate one image that covered a large area while preserving high image resolution at single cell level. To show inter-donor variability, data obtained from technical replicates were averaged to obtain the value of a given donor or condition.”

Please, let us know if the reviewer identifies any other detail that we should modify.

REVIEWER COMMENTS

Reviewer #1 (Remarks to the Author):

In general, this extensively revised manuscript addresses many issues raised during the initial review. In particular, the revisions have clarified numerous technical details missing from the initial submission, corrected several inaccurate or imprecise statements in the Introduction and Discussion, and now include experimental/biological replicates that address reproducibility of several key findings. In these respects, the manuscript is significantly improved.

The most exciting aspect of this study remains the novel MPS platform, which allows for the investigation of NK cell-tumor interactions in a 3D setting that mimics certain aspects of a bona fide tissue environment (e.g., NK cell extravasation and trafficking to the tumor, and interactions among multiple cell types). However, notwithstanding the technical innovation associated with the use of the MPS platform, it is unclear whether the key experimental findings represent major advances or are highly relevant to physiological responses in PLWH. For example, the findings on NK-tumor interactions in the MPS environment largely recapitulate, but do not necessarily advance upon, findings from prior studies in less complex in vitro culture system (e.g., ML-NKs are known to have different transcriptional profiles than naïve NK cells, and to be better killers; NK cells are known to kill HIV-infected T cells). Similarly, the finding that the presence of HIV-activated T cells enhances NK cell killing and extravasation/trafficking to the tumor is interesting, but only moderately impactful given the lack of novel mechanistic insights. And finally, it is unclear whether the finding that high numbers of newly infected CD4 T cells influence NK-tumor interactions in the MPS environment has direct relevance to NK-tumor interactions in immunocompromised HIV+ individuals with very low T cell counts.

Reviewer #2 (Remarks to the Author):

Issues have been addressed

Reviewer #3 (Remarks to the Author):

The authors have extensively revised the manuscript and in particular explained better as to what clinical therapy they are modeling in their studies. As this reviewer understands it, the study is modeling the use of ML-NK adoptive therapy in HIV+ patients. With this better understanding, I still have some concerns:

1. The abstract discusses the broad area of immunotherapy for cancer in HIV+ patients, and says (twice!), that HIV patients are usually excluded from such trials. The article would be improved if the abstract focuses on ML-NK adoptive therapy and how they are exploring how this may be used in HIV+ patients. Also, they still say that “and one of the first studies evaluating immunotherapy in general for people living with HIV and cancer”. This is not true; a number of such studies and even clinical trials have been done, including therapy with IL-12, cereblon-binding immunomodulators; and check-point inhibitors. And IL-2 was even used early in the AIDS pandemic.
2. In the introduction, they should make it clearer that they are describing an in vitro model

for ML-NK adoptive therapy in HIV+ patients. The introduction would benefit from a brief description of the status of ML-NK adoptive therapy for cancer or other settings... there are reports of this approach ...and less on immunotherapy in general for HIV patients.

3. In the section where they describe the production of ML-NK cells, the authors should define just how they define such cells. Are they defining them just as primary NK cells exposed to IL-12, IL-15, and IL-18? Or do they have other parameters? This should be clarified.

4. A minor point: The authors talk about the "CDC guidelines". These are guidelines on Clinicalinfo.HIV.gov, and in fact are the recommendations of DHHS and the NIH Office of AIDS Research. This should be corrected.

5. A number of the clinical statements are awkwardly stated. They should have the manuscript reviewed by someone with a good knowledge of clinical HIV disease.

REVIEWER COMMENTS

Reviewer #1 (Remarks to the Author):

In general, this extensively revised manuscript addresses many issues raised during the initial review. In particular, the revisions have clarified numerous technical details missing from the initial submission, corrected several inaccurate or imprecise statements in the Introduction and Discussion, and now include experimental/biological replicates that address reproducibility of several key findings. In these respects, the manuscript is significantly improved.

The most exciting aspect of this study remains the novel MPS platform, which allows for the investigation of NK cell-tumor interactions in a 3D setting that mimics certain aspects of a bona fide tissue environment (e.g., NK cell extravasation and trafficking to the tumor, and interactions among multiple cell types). However, notwithstanding the technical innovation associated with the use of the MPS platform, it is unclear whether the key experimental findings represent major advances or are highly relevant to physiological responses in PLWH. For example, the findings on NK-tumor interactions in the MPS environment largely recapitulate, but do not necessarily advance upon, findings from prior studies in less complex in vitro culture system (e.g., ML-NKs are known to have different transcriptional profiles than naïve NK cells, and to be better killers; NK cells are known to kill HIV-infected T cells). Similarly, the finding that the presence of HIV-activated T cells enhances NK cell killing and extravasation/trafficking to the tumor is interesting, but only moderately impactful given the lack of novel mechanistic insights. And finally, it is unclear whether the finding that high numbers of newly infected CD4 T cells influence NK-tumor interactions in the MPS environment has direct relevance to NK-tumor interactions in immunocompromised HIV+ individuals with very low T cell counts.

We thank the reviewer for their positive comments regarding the technical innovation and improvements in the manuscript.

Regarding the overall relevance of the manuscript. The reviewer is right that microphysiological models have been gaining traction during the last decade, and we believe they will probably continue to gain popularity in the following years. In December 2022, Biden's administration signed into law the FDA Modernization Act 2.0, which states that "non-animal or human biology-based test methods, such as cell-based assays or microphysiological systems" can be used as alternatives to animal models to support the initiation of clinical trials. Thus, we believe models like the this one are now, arguably, extremely relevant.

There were other general aspects that we discussed in the previous version of the manuscript that we think added some relevance, such as ML-NK cells leading to a decrease in HIV activation without necessarily killing HIV-infected cells. Similarly, the observation that ML-NK cells retained most of their cytotoxic capacity even in the complete absence of CD4 T cells would be relevant for PLWH with low CD4 count.

In summary, following the reviewers' comments in the previous round, we minimized the discussion around HIV treatment to focus mostly on the technological and cancer component of the manuscript. Overall, these points are still briefly mentioned in the text, but as the reviewer pointed out, these changes

may have reduced the apparent impact of the study. If the reviewers/editor think this should be included again, we will be more than happy to do it.

Reviewer #2 (Remarks to the Author):

Issues have been addressed.

We thank the reviewer for their support.

Reviewer #3 (Remarks to the Author):

The authors have extensively revised the manuscript and in particular explained better as to what clinical therapy they are modeling in their studies. As this reviewer understands it, the study is modeling the use of ML-NK adoptive therapy in HIV+ patients. With this better understanding, I still have some concerns:

1. The abstract discusses the broad area of immunotherapy for cancer in HIV+ patients, and says (twice!), that HIV patients are usually excluded from such trials. The article would be improved if the abstract focuses on ML-NK adoptive therapy and how they are exploring how this may be used in HIV+ patients. Also, they still say that “and one of the first studies evaluating immunotherapy in general for people living with HIV and cancer”. This is not true; a number of such studies and even clinical trials have been done, including therapy with IL-12, cereblon-binding immunomodulators; and check-point inhibitors. And IL-2 was even used early in the AIDS pandemic.

We agree with the reviewer that the abstract should focus primarily on adoptive therapy, with special emphasis on ML-NK cells. As the reviewer is pointing out, previous studies have evaluated immune checkpoint inhibitors in PLWH. Thus, we have modified the abstract following the reviewer’s suggestions to focus it on ML-NK cell adoptive therapy. Additionally, we have also modified the referenced sentences to focus the discussion on adoptive ML-NK cell therapy. Please, see new abstract below:

“Numerous studies are exploring the use of cell adoptive therapies to treat hematological malignancies as well as solid tumors. These therapies are based on infusions of immune cells such as T or Natural Killer (NK) cells to destroy tumor cells. However, there are numerous factors that dampen the immune response, leading to immunosuppression and potentially limiting the efficacy of adoptive cell therapy. This question is particularly important to the almost forty million people currently living with human immunodeficiency virus (HIV) worldwide. HIV infection leads to CD4 T cell depletion and immunosuppression, which in turn increases their risk of cancer and limits the capacity of the patient’s immune system to fight infections and tumor cells. While antiretroviral therapy extends life and suppresses HIV, the degree of immune recovery varies, leaving some people living with HIV (PLWH) in need of more efficient treatments against cancer. However, cancer in PLWH is difficult to mimic with traditional in vitro and in vivo systems in part because of variable immune recovery. In this study, we leveraged human-derived microphysiological models to reverse-engineer the HIV-immune system interaction and evaluated the potential of memory-like natural killer (ML-NK) cells for HIV+ head and neck cancer, one of the most common tumors in patients living with

HIV. ML-NK cells represent a subset of NK cells that derive from activated NK cells and exhibit increased cytotoxicity, faster activation, and elongated lifespan. This is the first report evaluating ML-NK cells for HIV+ patients with head and neck cancer. We evaluated multiple aspects of the ML-NK cell response in human-derived bioengineered HIV+ environments, including ML-NK cell extravasation, tumor penetration, tumor killing, T cell dependence, HIV suppression, and compatibility with anti-HIV medication. Overall, these results suggest that ML-NK cells are capable of operating without T cell assistance and could simultaneously destroy head and neck cancer cells as well as reduce HIV latency."

2. In the introduction, they should make it clearer that they are describing an in vitro model for ML-NK adoptive therapy in HIV+ patients. The introduction would benefit from a brief description of the status of ML-NK adoptive therapy for cancer or other settings... there are reports of this approach ...and less on immunotherapy in general for HIV patients.

Following the reviewer's suggestions, we have decreased the level of detail on HIV treatment/HIV immunotherapy and emphasized the sections about ML-NK cells. See below removed text about HIV immunotherapy:

~~"Multiple reviews have found that immune checkpoint inhibitors in HIV+ cancer patients are well tolerated, showing clinically relevant antitumor activity, and improving patient outcomes. Additionally, immune checkpoint inhibitors in HIV+ cancer patients are not associated with adverse changes in HIV viremia or CD4 T cell count. However, clinical trials exploring adoptive cell immunotherapy commonly still exclude immunocompromised individuals (NCT03296137, NCT04729543, NCT05451784, NCT03068819, NCT02782546). This situation is especially relevant for the almost forty million people living with HIV worldwide."~~

See below new text discussing clinical trials using ML-NK cells:

"In 2016, a first-in-human phase 1 clinical trial demonstrated that adoptive ML-NK cell therapy was feasible and safe in humans, leading to successful ML-NK cell proliferation and expansion in acute lymphoblastic leukemia patients²⁰. This study also showed that five out of nine patients developed robust anti-tumor response against AML cells, with four of them exhibiting complete remission²⁰. Another phase 1 clinical trial explored the potential of ML-NK cell therapy combined with chemotherapy in AML patients. The results showed that chemotherapy (i.e., fludarabine, cytarabine, and filgrastim) followed by ML NK cell therapy led to donor-derived ML-NK cell expansion for more than 3 months, and complete remission in four out of eight patients with no significant toxicity^{21, 22}. Other studies are exploring the use of single cell analysis techniques to develop molecular signatures (e.g., NKG2A) to identify and predict NK cell donors that will lead to optimal ML-NK cell response against tumor cells in humans²². As our knowledge of ML-NK cells advances, additional clinical trials continue to report promising results regarding the potential of ML-NK cells to treat hematological cancers^{23,24}, hoping to expand soon these findings to solid tumors²⁵."

3. In the section where they describe the production of ML-NK cells, the authors should define just how they define such cells. Are they defining them just as primary NK cells exposed to IL-12, IL-15, and IL-18? Or do they have other parameters? This should be clarified.

We have defined ML-NK cells as primary NK cells exposed to 10ng/mL IL-12, 50ng/mL IL-15, and 50ng/mL IL-18 for 16 hours and then cultured for a week in 1ng/mL IL-15. We have added a sentence making this explicit in the section where we describe the production of ML-NK cells for the first time:

“In this work, we generated ML-NK cells by exposing primary NK cells to 10ng/mL IL-12-, 50ng/mL IL-15, and 50ng/mL IL-18 for 16 hours followed by an additional week in culture in the presence of 1ng/mL IL-15.”

4. A minor point: The authors talk about the “CDC guidelines”. These are guidelines on Clinicalinfo.HIV.gov, and in fact are the recommendations of DHHS and the NIH Office of AIDS Research. This should be corrected.

Following the reviewer’s comment, we have modified the sentence. Additionally, following reviewer point #5 (see below), our colleague Dr. Striker reviewed these “clinical” statements to ensure they are phrased properly. Please see new sentence below:

“The standard of care for HIV patients today is a combination of at least two and more typically 3 antiretrovirals that have different mechanisms of action (e.g., integrase or protease inhibitors, and nucleoside or non-nucleoside reverse transcriptase inhibitors) ⁴⁵”.

5. A number of the clinical statements are awkwardly stated. They should have the manuscript reviewed by someone with a good knowledge of clinical HIV disease.

Following the reviewer’s suggestion, we have removed some statements about clinical HIV care that may have been convoluted and unnecessarily rigid about the evolving status of immunotherapy in immunocompromised patients. More specifically we have modified the abstract and shortened and simplified the paragraphs discussing HIV immunotherapy and treatment (see previous comments for more details). During the previous revision rounds, we incorporated our colleague Robert Striker to the author’s list. Dr. Striker has more than two decades of experience providing clinical care for PLWH and has contributed to revising the text and main findings and put them in context with HIV treatment in the clinic.

See below a few examples:

“The standard of care for HIV patients today is a combination of at least two and more typically 3 antivirals that have different mechanisms of action (integrase or protease inhibitors, and nucleoside or non-nucleoside reverse transcriptase inhibitors⁴²”.

“Previous studies using HIV-1 in animals (e.g., macaques, chimpanzees, gibbons) have provided valuable lessons about HIV biology²⁸ and SIV research in macaques has significantly advanced our knowledge²⁹.”

As discussed, we have removed or reduced several sections discussing HIV treatment. As per reviewer’s suggestions, see removed text below:

~~“Multiple reviews have found that immune checkpoint inhibitors in HIV+ cancer patients are well tolerated, showing clinically relevant antitumor activity, and improving patient outcomes. Additionally, immune checkpoint inhibitors in HIV+ cancer patients are not associated with adverse changes in HIV viremia or CD4 T cell count. However, clinical trials exploring adoptive cell immunotherapy commonly still exclude immunocompromised individuals (NCT03296137, NCT04729543, NCT05451784, NCT03068819, NCT02782546). This situation is especially relevant for the almost forty million people living with HIV worldwide.”~~

REVIEWERS' COMMENTS

Reviewer #1 (Remarks to the Author):

The authors have addressed most concerns raised in prior rounds of review. However, the fact that the key findings of the study largely confirm, but do not significantly advance, prior work remains a moderate concern. This concern is somewhat (but not fully) mitigated by the novelty and potential utility of the MPS platform.

Reviewer #3 (Remarks to the Author):

The authors have adequately addressed the concerns raised by this reviewer in the previous reviews. I believe this has substantially improved the manuscript